# One Step of Gradient Descent is Provably the Optimal In-Context Learner with One Layer of Linear Self-Attention

## Abstract

Recent works have empirically analyzed in-context learning and shown that transformers trained on synthetic linear regression tasks can learn to implement ridge regression, which is the Bayes-optimal predictor, given sufficient capacity (Akyürek et al., 2023), while one-layer transformers with linear self-attention and no MLP layer will learn to implement one step of gradient descent (GD) on a least-squares linear regression objective (von Oswald et al., 2022). However, the theory behind these observations remains poorly understood. We theoretically study transformers with a single layer of linear self-attention, trained on synthetic noisy linear regression data. First, we mathematically show that when the covariates are drawn from a standard Gaussian distribution, the one-layer transformer which minimizes the pre-training loss will implement a single step of GD on the least-squares linear regression objective. Then, we find that changing the distribution of the covariates and weight vector to a non-isotropic Gaussian distribution has a strong impact on the learned algorithm: the global minimizer of the pre-training loss now implements a single step of *pre-conditioned* GD. However, if only the distribution of the responses is changed, then this does not have a large effect on the learned algorithm: even when the response comes from a more general family of *nonlinear* functions, the global minimizer of the pre-training loss still implements a single step of GD on a least-squares linear regression objective.

## 1 Introduction

Large language models (LLMs) demonstrate the surprising ability of in-context learning, where an LLM "learns" to solve a task by conditioning on a prompt containing input-output exemplars (Brown et al., 2020; Lieber et al., 2021; Radford et al., 2019; Wang & Komatsuzaki, 2021). Recent works have advanced the understanding of in-context learning via empirical analysis (Min et al., 2022; Wei et al., 2023; Akyürek et al., 2023; von Oswald et al., 2022; Dai et al., 2023), but theoretical analysis remains limited (Xie et al., 2022).

A recent line of work (Garg et al., 2022; Akyürek et al., 2023; von Oswald et al., 2022; Dai et al., 2023) empirically finds that transformers can be trained to implement algorithms that solve linear regression problems in-context. Specifically, in each input sequence the transformer is given a set of in-context examples $(x_i, y_i)$, where $y_i = w^\top x_i + \epsilon_i$ with a shared and hidden random coefficient vector $w$ and random noise $\epsilon_i$, and a test example $x$.[1] The transformer is then trained to predict $y = w^\top x + \epsilon$, where $\epsilon$ denotes random noise from the same distribution as $\epsilon_i$. These works find that the transformer outputs a prediction $\hat{y}$ which is similar to the predictions of existing, interpretable linear regression algorithms, such as gradient descent (GD) or ordinary least squares, applied to the dataset consisting of the pairs $(x_i, y_i)$. In particular, von Oswald et al. (2022) empirically show that a one-layer transformer with linear self-attention and no MLP layer will implement a single step of gradient descent when trained on such a distribution.

Several works (e.g. Akyürek et al. (2023); Liu et al. (2023); Giannou et al. (2023)) theoretically study the expressive power of transformers. In the context of linear regression tasks, Akyürek et al.

---

[1] In some settings in these works, the noise is set to 0.

(2023) describe how transformers can represent gradient descent, or Sherman-Morrison updates, and Giannou et al. (2023) describe how transformers can represent Newton's algorithm for matrix inversion. However, in addition to the expressive power of transformers, it is also of interest to understand the behavior of transformers trained with gradient-based algorithms. Furthermore, it is still useful to understand the behavior of models with restricted capacity—though practical LLMs are very expressive, they need to perform many tasks simultaneously, and therefore the capacity per problem may still be relatively limited. Thus, motivated by von Oswald et al. (2022), we theoretically study the global minima of the pre-training loss for one-layer transformers with linear self-attention on the linear regression data distribution described above.

**Contributions.** In this paper, we study transformers with one linear self-attention layer, and mathematically investigate which algorithms the transformers implement for synthetically generated linear regression datasets. We prove that the transformer which implements a single step of gradient descent on a least squares linear regression objective is the global minimizer of the pre-training loss. This exactly matches the empirical findings of von Oswald et al. (2022).

Concretely, we consider a setup similar to von Oswald et al. (2022); Akyürek et al. (2023). The model we study is a transformer with one linear single-head self-attention layer, which is the same model as the one empirically studied by von Oswald et al. (2022). The training data for this transformer consist of sequences of the form $(x_1, y_1, \ldots, x_n, y_n)$, where the $x_i$ are sampled from $\mathcal{N}(0, I_{d \times d})$ and $y_i = w^\top x_i + \epsilon_i$, where $w$ is sampled from $\mathcal{N}(0, I_{d \times d})$ once per sequence, and the $\epsilon_i$ are i.i.d. Gaussian noise with variance $\sigma^2$. The pre-training loss is the expected error that the transformer achieves when predicting $y = w^\top x$ given the test example $x$ and the context $(x_1, y_1, \ldots, x_n, y_n)$, i.e. the pre-training loss is $L = \mathbb{E}_{(x_1, y_1, \ldots, x_n, y_n), x, y}[(y - \hat{y})^2]$, where $\hat{y}$ is the output of the transformer given $(x_1, y_1, \ldots, x_n, y_n)$ and $x$ as input.

We show in Section 3 that the transformer which is the global minimizer of the pre-training loss $L$ implements one step of gradient descent on a linear regression objective with the dataset consisting of the $(x_i, y_i)$. More concretely, the transformer implements the prediction algorithm

$$\hat{y} = \eta \sum_{i=1}^n y_i x_i^\top x \,. \tag{1}$$

where $\eta$ is a learning rate independent of the $x_i$, the $y_i$ and $x$. However, one step of GD is also preferred in part due to the distribution of the $x_i$. In particular, if the covariance of $x_i$ is no longer the identity matrix, we show (Section 4) that the global minimum of the pre-training loss corresponds to one step of GD, but with pre-conditioning.

Interestingly, our theory also suggests that the distribution of $y_i | x_i$ does not play such a significant role in the algorithm learned by the transformer. In Section 5, we study a setting where $y_i | x_i$ is *nonlinear*, but satisfies some mild assumptions, such as invariance to rotations of the distribution of the $x_i$. As a concrete special case, the target function can be a neural network with any depth/width and i.i.d. random Gaussian weights. We show in Section 5 that a one-layer transformer with linear self-attention, which minimizes the pre-training loss, still implements one step of GD on a **linear regression** objective. Intuitively, this is likely because of the constraint imposed by the architecture, which prevents the transformer from making use of any more complex structure in the $y_i$.

**Concurrent Works.** We discuss the closely related works of Ahn et al. (2023) and Zhang et al. (2023) which are concurrent with and independent with our work and were posted prior to our work on arXiv. Ahn et al. (2023) give theoretical results very similar to ours. They study one-layer transformers with linear self-attention with the same parameterization as von Oswald et al. (2022), and show that with isotropic $x_i$, the global minimizer of the pre-training loss corresponds to one step of gradient descent on a linear model. They also show that for more general covariance matrices, the global minimizer of the pre-training loss corresponds to one step of pre-conditioned gradient descent, where the pre-conditioner matrix can be computed in terms of the covariance of $x_i$.[2]

Different from our work, Ahn et al. (2023) also show additional results for multi-layer transformers (with linear self-attention) with residual connections trained on linear regression data. First, they study a restricted parameterization where in each layer, the product of the projection and value

---

[2]This result is not exactly the same as our result in Section 4, since we assume $w \sim \mathcal{N}(0, \Sigma^{-1})$ while they assume $w \sim \mathcal{N}(0, I_{d \times d})$.

matrices has only one nonzero entry. In this setting, for two-layer transformers with linear self-attention, they show that the global minimizer corresponds to two steps of pre-conditioned GD with diagonal pre-conditioner matrices, when the data is isotropic. For linear transformers with $k$ layers, they show that $k$ steps of pre-conditioned GD corresponds to a critical point of the pre-training loss,[3] where the pre-conditioner matrix is the inverse of the covariance matrix of the $x_i$.[4] Next, they study a less restrictive parameterization where the product of the projection and value matrices can be almost fully dense, and show that a certain critical point of the pre-training loss for $k$-layer linear transformers corresponds to $k$ steps of a generalized version of the GD++ algorithm, which was empirically observed by von Oswald et al. (2022) to be the algorithm learned by $k$-layer linear transformers.

Zhang et al. (2023) also theoretically study a setting similar to ours. They not only show that the global minimizer of the pre-training loss implements one step of GD (the same result as ours), but also show that a one-layer linear transformer trained with gradient flow will converge to a global minimizer. They also show that the transformer implements a step of pre-conditioned GD when the $x_i$ are non-isotropic. They also characterize how the training prompt lengths and test prompt length affect the test-time prediction error of the trained transformer. Additionally, they consider the behavior of the trained transformer under distribution shifts, as well as the training dynamics when the covariance matrices of the $x_i$ in different training prompts can be different.

One additional contribution of our work is that we also consider the case where the target function in the pre-training data is not a linear function (Section 5). This suggests that, compared to the distribution of the covariates, the distribution of the responses at training time does not have as strong of an effect on the algorithm learned by the transformer. We note that our proof in this setting is not too different from our proof in Section 3. Zhang et al. (2023) consider the case where the $y_i$'s in the test time prompt are obtained from a nonlinear target function, and consider the performance on this prompt of the transformer trained on prompts with a linear target function — this is different from our setting in Section 5 since we consider the case where the training prompts themselves are obtained with a nonlinear target function. We discuss additional related works in Appendix A.

## 2 SETUP

Our setup is similar to von Oswald et al. (2022).

**One-Layer Transformer with Linear Self-Attention.** A linear self-attention layer with width $s$ consists of the following parameters: a key matrix $W_K \in \mathbb{R}^{s \times s}$, a query matrix $W_Q \in \mathbb{R}^{s \times s}$, and a value matrix $W_V \in \mathbb{R}^{s \times s}$. Given a sequence of $T > 1$ tokens $(v_1, v_2, \ldots, v_T)$, the output of the linear self-attention layer is defined to be $(\hat{v}_1, \hat{v}_2, \ldots, \hat{v}_T)$, where for $i \in [T]$ with $i > 1$,

$$\hat{v}_i = \sum_{j=1}^{i-1} (W_V v_j)(v_j^\top W_K^\top W_Q v_i), \tag{2}$$

and $\hat{v}_1 = 0$. In particular, the output on the $T^{\text{th}}$ token is

$$\hat{v}_T = \sum_{j=1}^{T-1} (W_V v_j)(v_j^\top W_K^\top W_Q v_T). \tag{3}$$

As in the theoretical construction of von Oswald et al. (2022), we do not consider the attention score between a token $v_i$ and itself. Our overall transformer is then defined to be a linear self-attention layer with key matrix $W_K$, query matrix $W_Q$, and value matrix $W_V$, together with a linear head $h \in \mathbb{R}^s$ which is applied to the last token. Thus, the final output of the transformer is $h^\top \hat{v}_T$. We will later instantiate this one-layer transformer with $s = d + 1$, where $d$ is the dimension of the inputs $x_i$. We note that this corresponds to a single head of linear self-attention, while one could also consider multi-head self-attention.

---

[3]One technical point is that they show there exist transformers representing this form of pre-conditioned GD having arbitrarily small gradient, but not that there exists a transformer with gradient exactly 0 which represents this form of pre-conditioned GD.

[4]Here, they assume that $x_i \sim \mathcal{N}(0, \Sigma)$ and $w \sim \mathcal{N}(0, \Sigma^{-1})$, which is the same assumption as our result in Section 4 but different from their result for one-layer transformers where $x_i \sim \mathcal{N}(0, \Sigma)$.

**Linear Regression Data Distribution.** The pretraining data distribution consists of sequences $D = (x_1, y_1, \ldots, x_{n+1}, y_{n+1})$. Here, the *exemplars* $x_i$ are sampled i.i.d. from $\mathcal{N}(0, I_{d \times d})$. Then, a weight vector $w \in \mathbb{R}^d$ is sampled from $\mathcal{N}(0, I_{d \times d})$, freshly for each sequence. Finally, $y_i$ is computed as $y_i = w^\top x_i + \epsilon_i$ where $\epsilon_i \sim \mathcal{N}(0, \sigma^2)$ for some $\sigma > 0$. We consider the vector $v_i = \begin{bmatrix} x_i \\ y_i \end{bmatrix} \in \mathbb{R}^{d+1}$ to be a *token* — in other words, the sequence $(x_1, y_1, \ldots, x_{n+1}, y_{n+1})$ is considered to have $n+1$ tokens (rather than $2(n+1)$ tokens). We use $\mathcal{T}$ to denote the distribution of sequences defined in this way.

At both training and test time, $(x_1, y_1, \ldots, x_n, y_n, x_{n+1}, y_{n+1})$ is generated according to the pre-training distribution $\mathcal{T}$, i.e. the $x_i$ are sampled i.i.d. from $\mathcal{N}(0, I_{d \times d})$, a new weight vector $w \in \mathbb{R}^d$ is also sampled from $\mathcal{N}(0, I_{d \times d})$, and $y_i = w^\top x_i + \epsilon_i$ where the $\epsilon_i$ are sampled i.i.d. from $\mathcal{N}(0, \sigma^2)$. Then, the in-context learner is presented with $x_1, y_1, \ldots, x_n, y_n, x_{n+1}$, and must predict $y_{n+1}$. We refer to $x_1, \ldots, x_n$ as the *support* exemplars and $x_{n+1}$ as the *query* exemplar. Here, $v_1, \ldots, v_n$ are defined as above, but $v_{n+1} = \begin{bmatrix} x_{n+1} \\ 0 \end{bmatrix}$, following the notation of von Oswald et al. (2022).[5] We note that this is not significantly different from the standard in-context learning setting, since even though the final token $v_{n+1}$ has 0 as an extra coordinate, it does not provide the transformer with any additional information about $y_{n+1}$.

**Loss Function.** Given a one-layer transformer with linear self-attention and width $d + 1$, with key matrix $W_K \in \mathbb{R}^{(d+1) \times (d+1)}$, query matrix $W_Q \in \mathbb{R}^{(d+1) \times (d+1)}$, and value matrix $W_V \in \mathbb{R}^{(d+1) \times (d+1)}$, and with a head $h \in \mathbb{R}^{d+1}$, the loss of this transformer on our linear regression data distribution is formally defined as

$$L(W_K, W_Q, W_V, h) = \mathbb{E}_{D \sim \mathcal{T}}[(h^\top \hat{v}_{n+1} - y_{n+1})^2], \tag{4}$$

where as defined above, $\hat{v}_{n+1}$ is the output of the linear self-attention layer on the $(n+1)^{\text{th}}$ token, which in this case is $\begin{bmatrix} x_{n+1} \\ 0 \end{bmatrix}$.

We now rewrite the loss function and one-layer transformer in a more convenient form. As a convenient shorthand, for any test-time sequence $D = (x_1, y_1, \ldots, x_{n+1}, 0)$, we write $\widetilde{D} = (x_1, y_1, \ldots, x_n, y_n)$, i.e. the prefix of $D$ that does not include $x_{n+1}$ and $y_{n+1}$. We also define

$$G_{\widetilde{D}} = \sum_{i=1}^{n} \begin{bmatrix} x_i \\ y_i \end{bmatrix} \begin{bmatrix} x_i \\ y_i \end{bmatrix}^\top. \tag{5}$$

With this notation, we can write the prediction obtained from the transformer on the final token as

$$\hat{y}_{n+1} = h^\top W_V G_{\widetilde{D}} W_K^\top W_Q v_{n+1}. \tag{6}$$

where $v_{n+1} = \begin{bmatrix} x_{n+1} \\ 0 \end{bmatrix}$. Additionally, we also define the matrix $X \in \mathbb{R}^{n \times d}$ as the matrix whose $i^{th}$ row is the row vector $x_i^\top$, i.e.

$$X = \begin{bmatrix} \cdots & x_1^\top & \cdots \\ \cdots & x_2^\top & \cdots \\ \vdots & \vdots & \vdots \\ \cdots & x_n^\top & \cdots \end{bmatrix}, \tag{7}$$

and we define the vector $\vec{y} \in \mathbb{R}^n$ as the vector whose $i^{\text{th}}$ entry is $y_i$, i.e.

$$\vec{y} = \begin{bmatrix} y_1 \\ y_2 \\ \vdots \\ y_n \end{bmatrix}. \tag{8}$$

---

[5]If we were to treat $x_i$ and $y_i$ as separate tokens, then we would need to deal with attention scores between $y_i$ and $y_j$ for $i \neq j$, as well as attention scores between $y_i$ and $x_j$ for $i \neq j$. Our current setup simplifies the analysis.

Finally, it is worth noting that we can write the loss function as

$$L(W_K, W_Q, W_V, h) = \mathbb{E}_{D \sim \mathcal{T}}[(h^\top W_V G_{\widetilde{D}} W_K^\top W_Q v_{n+1} - y_{n+1})^2]. \qquad (9)$$

Thus, for $w \in \mathbb{R}^{d+1}$ and $M \in \mathbb{R}^{(d+1) \times (d+1)}$, if we define

$$L(w, M) = \mathbb{E}_{D \sim \mathcal{T}}[(w^\top G_{\widetilde{D}} M v_{n+1} - y_{n+1})^2], \qquad (10)$$

then $L(W_V^\top h, W_K^\top W_Q) = L(W_K, W_Q, W_V, h)$. Note that we have a slight abuse of notation, and $L$ has two different meanings depending on the number of arguments. Finally, with the change of variables $M = W_K^\top W_Q$ and $w = W_V^\top h$, we can write the prediction of the transformer as $w^\top G_{\widetilde{D}} M v_{n+1}$. Thus, the output of the transformer only depends on the parameters through $w^\top G_{\widetilde{D}} M$.

**Additional Notation.** For a matrix $A \in \mathbb{R}^{d \times d}$, we write $A_{i:j,:}$ to denote the sub-matrix of $A$ that contains the rows of $A$ with indices between $i$ and $j$ (inclusive). Similarly, we write $A_{:,i:j}$ to denote the sub-matrix of $A$ that contains the columns of $A$ with indices between $i$ and $j$ (inclusive). We write $A_{i:j,k:l}$ to denote the sub-matrix of $A$ containing the entries with row indices between $i$ and $j$ (inclusive) and column indices between $k$ and $l$ (inclusive).

## 3 MAIN RESULT FOR LINEAR MODELS

**Theorem 1** (Global Minimum for Linear Regression Data). *Suppose $(W_K^*, W_Q^*, W_V^*, h^*)$ is a global minimizer of the loss $L$. Then, the corresponding one-layer transformer with linear self-attention implements one step of gradient descent on a linear model with some learning rate $\eta > 0$. More concretely, given a query token $v_{n+1} = \begin{bmatrix} x_{n+1} \\ 0 \end{bmatrix}$, the transformer outputs $\eta \sum_{i=1}^n y_i x_i^\top x_{n+1}$, where $\eta = \frac{\mathbb{E}_{\widetilde{D} \sim \mathcal{T}}[\hat{w}_{\widetilde{D}}^\top X^\top \vec{y}]}{\mathbb{E}_{\widetilde{D} \sim \mathcal{T}}[\vec{y}^\top X X^\top \vec{y}]}$. Here given a prefix $\widetilde{D}$ of a test-time data sequence $D$, we let $\hat{w}_{\widetilde{D}}$ denote the solution to ridge regression on $X$ and $\vec{y}$ with regularization strength $\sigma^2$.*

The minimizer $(W_K^*, W_Q^*, W_V^*, h^*)$ is not unique, though the linear predictor implemented by the minimizer is unique — see the discussion after Lemma 2. One such construction is as follows. von Oswald et al. (2022) describe essentially the same construction, but our result shows that it is a global minimum of the loss function, while von Oswald et al. (2022) do not theoretically study the construction aside from showing that it is equivalent to one step of gradient descent. We define

$$W_K^* = \begin{pmatrix} I_{d \times d} & 0 \\ 0 & 0 \end{pmatrix}, W_Q^* = \begin{pmatrix} I_{d \times d} & 0 \\ 0 & 0 \end{pmatrix}, W_V^* = \begin{pmatrix} 0 & 0 \\ 0 & \eta \end{pmatrix}, h^* = \begin{bmatrix} 0 \\ 1 \end{bmatrix}. \qquad (11)$$

Here, the unique value of $\eta$ which makes this construction a global minimum is $\eta = \frac{\mathbb{E}_{\widetilde{D} \sim \mathcal{T}}[\hat{w}_{\widetilde{D}}^\top X^\top \vec{y}]}{\mathbb{E}_{\widetilde{D} \sim \mathcal{T}}[\vec{y}^\top X X^\top \vec{y}]}$. To see why this construction implements a single step of gradient descent on a linear model, note that given test time inputs $x_1, y_1, \ldots, x_n, y_n, x_{n+1}$, if we write $v_i = \begin{bmatrix} x_i \\ y_i \end{bmatrix}$ for $i \leq n$ and $v_{n+1} = \begin{bmatrix} x_{n+1} \\ 0 \end{bmatrix}$, then the output of the corresponding transformer would be

$$(h^*)^\top \sum_{i=1}^n (W_V^* v_i)(v_i^\top (W_K^*)^\top W_Q^* v_{n+1}) = \eta \sum_{i=1}^n y_i x_i^\top x_{n+1}. \qquad (12)$$

On the other hand, consider linear regression with total squared error as the loss function, using the $x_i$ and $y_i$. Here, the loss function would be $L(w) = \frac{1}{2} \sum_{i=1}^n (w^\top x_i - y_i)^2$, meaning that the gradient is $\nabla_w L(w) = \sum_{i=1}^n (w^\top x_i - y_i) x_i$. In particular, if we initialize gradient descent at $w_0 = 0$, then after one step of gradient descent with learning rate $\eta$, the iterate would be at $w_1 = \eta \sum_{i=1}^n y_i x_i$ — observe that the final expression in Equation (12) is exactly $w_1^\top x_{n+1}$.

Now, we give an overview of the proof of Theorem 1. By the discussion in Section 2, it suffices to show that $L((W_V^*)^\top h^*, (W_K^*)^\top W_Q^*)$ is the global minimum of $L(w, M)$. The first step of the proof is to rewrite the loss in a more convenient form, getting rid of the expectation over $x_{n+1}$ and $y_{n+1}$:

**Lemma 1.** *Let $\hat{w}_{\widetilde{D}}$ be the solution to ridge regression with regularization strength $\sigma^2$ on the exemplars $(x_1, y_1), \ldots, (x_n, y_n)$ given in a context $\widetilde{D}$. Then, there exists a constant $C \geq 0$, which is independent of $w, M$, such that $L(w, M) = C + \mathbb{E}_{D \sim \mathcal{T}} \|M_{:,1:d}^\top G_{\widetilde{D}} w - \hat{w}_{\widetilde{D}}\|_2^2$.*

As discussed towards the end of Section 2, the prediction can be written as $w^\top G_{\widetilde{D}} M v_{n+1}$ where $v_{n+1} = \begin{bmatrix} x_{n+1} \\ 0 \end{bmatrix}$, meaning that the effective linear predictor implemented by the transformer is the linear function from $\mathbb{R}^d$ to $\mathbb{R}$ with weight vector $M_{:,1:d}^\top G_{\widetilde{D}} w$. Thus, we can interpret Lemma 1 as saying that the loss function encourages the effective linear predictor to match the Bayes-optimal predictor $\hat{w}_{\widetilde{D}}$. Note that it is not possible for the effective linear predictor of the transformer to match $\hat{w}_{\widetilde{D}}$ exactly, since the transformer can only implement a linear or quadratic function of the $x_i$, while representing $\hat{w}_{\widetilde{D}}$ requires computing $(X^\top X + \sigma^2 I)^{-1}$, and this is a much more complex function of the $x_i$. We prove Lemma 1 using the fact that $\mathbb{E}_{\widetilde{D}, x_{n+1}}[y_{n+1}] = \hat{w}_{\widetilde{D}}^\top x_{n+1}$ and standard manipulations of random variables — we give a detailed proof in Appendix C.

Next, the key step is to replace $\hat{w}_{\widetilde{D}}$ in the above lemma by $\eta X^\top \vec{y}$.

**Lemma 2.** *There exists a constant $C_1 \geq 0$ which is independent of $w, M$, such that*

$$L(w, M) = C_1 + \mathbb{E}_{\widetilde{D} \sim \mathcal{T}} \|M_{:,1:d}^\top G_{\widetilde{D}} w - \eta X^\top \vec{y}\|_2^2. \tag{13}$$

Lemma 2 says that the loss depends entirely on how far the effective linear predictor is from $\eta X^\top \vec{y}$. It immediately follows from this lemma that $(W_K, W_Q, W_V, h)$ is a global minimizer of the loss if and only if the effective linear predictor of the corresponding transformer is $\eta X^\top \vec{y}$. Thus, Theorem 1 follows almost directly from Lemma 2, and in the rest of this section, we give an outline of the proof of Lemma 2 — the detailed proofs of Theorem 1 and Lemma 2 are in Appendix C.

**Proof Strategy for Lemma 2.** Our overall proof strategy is to show that the gradients of $L(w, M)$ and $L'(w, M)$, defined as

$$L(w, M) := \mathbb{E}_{\widetilde{D} \sim \mathcal{T}} \|M_{:,1:d}^\top G_{\widetilde{D}} w - \hat{w}_{\widetilde{D}}\|_2^2, \quad L'(w, M) := \mathbb{E}_{\widetilde{D} \sim \mathcal{T}} \|M_{:,1:d}^\top G_{\widetilde{D}} w - \eta X^\top \vec{y}\|_2^2, \tag{14}$$

are equal at every $w, M$, from which Lemma 2 immediately follows.[6] For simplicity, we write $A = M_{:,1:d}^\top$, so without loss of generality, we can instead show that the gradients of the loss functions $J_1(A, w)$ and $J_2(A, w)$ are identical, where $J_1$ and $J_2$ are defined as

$$J_1(A, w) := \mathbb{E}_{\widetilde{D} \sim \mathcal{T}} \|AG_{\widetilde{D}} w - \hat{w}_{\widetilde{D}}\|_2^2, \quad J_2(A, w) := \mathbb{E}_{\widetilde{D} \sim \mathcal{T}} \|AG_{\widetilde{D}} w - \eta X^\top \vec{y}\|_2^2. \tag{15}$$

In this section, we discuss the gradients with respect to $w$ — we use the same proof ideas to show that the gradients with respect to $A$ are the same. We have

$$\nabla_w J_1(A, w) = 2\mathbb{E}_{\widetilde{D} \sim \mathcal{T}} G_{\widetilde{D}} A^\top (AG_{\widetilde{D}} w - \hat{w}_{\widetilde{D}}) \tag{16}$$

and

$$\nabla_w J_2(A, w) = 2\mathbb{E}_{\widetilde{D} \sim \mathcal{T}} G_{\widetilde{D}} A^\top (AG_{\widetilde{D}} w - \eta X^\top \vec{y}). \tag{17}$$

Thus, showing that these two gradients are equal for all $A, w$ reduces to showing that for all $A$, we have

$$\mathbb{E}_{\widetilde{D} \sim \mathcal{T}} G_{\widetilde{D}} A^\top \hat{w}_{\widetilde{D}} = \eta \mathbb{E}_{\widetilde{D} \sim \mathcal{T}} G_{\widetilde{D}} A^\top X^\top \vec{y}. \tag{18}$$

Recall that $G_{\widetilde{D}}$ is defined as $G_{\widetilde{D}} = \sum_{i=1}^n \begin{bmatrix} x_i x_i^\top & y_i x_i \\ y_i x_i^\top & y_i^2 \end{bmatrix}$. Our first key observation is that for any $i \in [n]$ and any odd positive integer $k$, $\mathbb{E}[y_i^k \mid X] = 0$, since $y_i = w^\top x_i + \epsilon_i$, and both $w^\top x_i$ and $\epsilon_i$ have distributions which are symmetric around 0. This observation also extends to any odd-degree

---

[6]This is because $L$ and $L'$ are defined everywhere on $\mathbb{R}^{d+1} \times \mathbb{R}^{(d+1) \times (d+1)}$, and for any two differentiable functions $f, g$ defined on an open connected subset $S \subset \mathbb{R}^k$, if the gradients of $f$ and $g$ are identical on $S$, then $f$ and $g$ are equal on $S$ up to an additive constant. This can be shown using the fundamental theorem of calculus.

monomial of the $y_i$. Using this observation, we can simplify the left-hand side of Equation (18) as follows. We can first write it as

$$\mathbb{E}_{\widetilde{D}\sim\mathcal{T}}G_{\widetilde{D}}A^\top\hat{w}_{\widetilde{D}} = \mathbb{E}_{\widetilde{D}\sim\mathcal{T}}\sum_{i=1}^n \left[ \begin{array}{c} x_i x_i^\top (A^\top)_{1:d,:}\hat{w}_{\widetilde{D}} + y_i x_i (A^\top)_{d+1,:}\hat{w}_{\widetilde{D}} \\ y_i x_i^\top (A^\top)_{1:d,:}\hat{w}_{\widetilde{D}} + y_i^2 (A^\top)_{d+1,:}\hat{w}_{\widetilde{D}} \end{array} \right]. \tag{19}$$

Then, since $\hat{w}_{\widetilde{D}} = (X^\top X + \sigma^2 I)^{-1}X^\top \vec{y}$, each entry of $\hat{w}_{\widetilde{D}}$ has an odd degree in the $y_i$, meaning the terms $x_i x_i^\top (A^\top)_{1:d,:}\hat{w}_{\widetilde{D}}$ and $y_i^2 (A^\top)_{d+1,:}\hat{w}_{\widetilde{D}}$ in the above equation vanish after taking the expectation. Thus, we obtain

$$\mathbb{E}_{\widetilde{D}\sim\mathcal{T}}G_{\widetilde{D}}A^\top\hat{w}_{\widetilde{D}} = \mathbb{E}_{\widetilde{D}\sim\mathcal{T}}\sum_{i=1}^n \left[ \begin{array}{c} y_i x_i A^\top_{:,d+1}\hat{w}_{\widetilde{D}} \\ y_i x_i^\top A^\top_{:,1:d}\hat{w}_{\widetilde{D}} \end{array} \right] = \mathbb{E}_{\widetilde{D}\sim\mathcal{T}} \left[ \begin{array}{c} X^\top\vec{y}A^\top_{:,d+1}\hat{w}_{\widetilde{D}} \\ \vec{y}^\top X A^\top_{:,1:d}\hat{w}_{\widetilde{D}} \end{array} \right]. \tag{20}$$

Since each entry of $\eta X^\top\vec{y}$ has an odd degree in the $y_i$, in order to simplify the right-hand side of Equation (18), we can apply the same argument but with $\hat{w}_{\widetilde{D}}$ replaced by $\eta X^\top\vec{y}$, obtaining

$$\eta\mathbb{E}_{\widetilde{D}\sim\mathcal{T}}G_{\widetilde{D}}A^\top X^\top\vec{y} = \eta\mathbb{E}_{\widetilde{D}\sim\mathcal{T}} \left[ \begin{array}{c} X^\top\vec{y}A^\top_{:,d+1}X^\top\vec{y} \\ \vec{y}^\top X A^\top_{:,1:d}X^\top\vec{y} \end{array} \right]. \tag{21}$$

Thus, showing Equation (18) reduces to showing that

$$\mathbb{E}_{\widetilde{D}\sim\mathcal{T}} \left[ \begin{array}{c} X^\top\vec{y}A^\top_{:,d+1}\hat{w}_{\widetilde{D}} \\ \vec{y}^\top X A^\top_{:,1:d}\hat{w}_{\widetilde{D}} \end{array} \right] = \eta\mathbb{E}_{\widetilde{D}\sim\mathcal{T}} \left[ \begin{array}{c} X^\top\vec{y}A^\top_{:,d+1}X^\top\vec{y} \\ \vec{y}^\top X A^\top_{:,1:d}X^\top\vec{y} \end{array} \right]. \tag{22}$$

To show Equation (22), our key tool is Lemma 4, which follows from Lemma 3.

**Lemma 3.** *There exists a scalar $c_1$ such that $\mathbb{E}_{\widetilde{D}\sim\mathcal{T}}[X^\top\vec{y}\vec{y}^\top X] = c_1 I_{d\times d}$, and there exists a scalar $c_2$ such that $\mathbb{E}_{\widetilde{D}\sim\mathcal{T}}[X^\top\vec{y}\hat{w}_{\widetilde{D}}^\top] = c_2 I_{d\times d}$.*

**Lemma 4.** *If $\eta = \frac{\mathbb{E}_{\widetilde{D}\sim\mathcal{T}}[\hat{w}_{\widetilde{D}}^\top X^\top\vec{y}]}{\mathbb{E}_{\widetilde{D}\sim\mathcal{T}}[\vec{y}^\top XX^\top\vec{y}]}$, then $\mathbb{E}_{\widetilde{D}\sim\mathcal{T}}[\eta X^\top\vec{y}\vec{y}^\top X - X^\top\vec{y}\hat{w}_{\widetilde{D}}^\top] = 0$.*

**Overview of Proof of Lemma 3 and Lemma 4.** We give an overview of how we prove Lemma 3 and Lemma 4 here, and defer the full proofs to Appendix C. To show that $\mathbb{E}_{\widetilde{D}\sim\mathcal{T}}[X^\top\vec{y}\vec{y}^\top X]$ is a scalar multiple of the identity, we first use the fact that even when all of the $x_i$ are rotated by a rotation matrix $R$, the distribution of $\vec{y}|X$ remains the same, since the weight vector $w$ is drawn from $\mathcal{N}(0, I_{d\times d})$ which is a rotationally invariant distribution. Thus, if we define $M(X) = \mathbb{E}[\vec{y}\vec{y}^\top \mid X]$ as a function of $X \in \mathbb{R}^{n\times d}$, then for any rotation matrix $R \in \mathbb{R}^{d\times d}$, we have

$$M(XR^\top) = \mathbb{E}[\vec{y}\vec{y}^\top \mid XR^\top] = \mathbb{E}[\vec{y}\vec{y}^\top \mid X] = M(X), \tag{23}$$

where the second equality is because multiplying $X$ on the right by $R^\top$ corresponds to rotating each of the $x_i$ by $R$. Additionally, if we rotate the $x_i$ by $R$, then $\mathbb{E}_{\widetilde{D}\sim\mathcal{T}}[X^\top\vec{y}\vec{y}^\top X]$ remains the same — this is because the distribution of the $x_i$ is unchanged due to the rotational invariance of the Gaussian distribution, and the conditional distribution $y_i \mid x_i$ is unchanged when we rotate $x_i$ by $R$. This implies that

$$\mathbb{E}[X^\top\vec{y}\vec{y}^\top X] = \mathbb{E}[X^\top M(X)X] = \mathbb{E}[(XR^\top)^\top M(XR^\top)XR^\top], \tag{24}$$

where the second equality is because, as we observed above, $\mathbb{E}_{\widetilde{D}\sim\mathcal{T}}[X^\top\vec{y}\vec{y}^\top X]$ remains the same when we rotate each of the $x_i$ by $R$. Since $M(XR^\top) = M(X)$, we have

$$\mathbb{E}[(XR^\top)^\top M(XR^\top)XR^\top] = R\mathbb{E}[X^\top M(X)X]R^\top, \tag{25}$$

which implies that $\mathbb{E}[X^\top\vec{y}\vec{y}^\top X] = R\mathbb{E}[X^\top\vec{y}\vec{y}^\top X]R^\top$ for any rotation matrix $R$, and therefore $\mathbb{E}[X^\top\vec{y}\vec{y}^\top X]$ is a scalar multiple of the identity matrix. To finish the proof of Lemma 3, we show that $\mathbb{E}_{\widetilde{D}\sim\mathcal{T}}[X^\top\vec{y}\hat{w}_{\widetilde{D}}^\top]$ is a scalar multiple of the identity using essentially the same argument. To show Lemma 4, we simply take the trace of $\mathbb{E}_{\widetilde{D}\sim\mathcal{T}}[\eta X^\top\vec{y}\vec{y}^\top X - X^\top\vec{y}\hat{w}_{\widetilde{D}}^\top]$, and select $\eta$ so that this trace is equal to 0.

**Finishing the Proof of Lemma 2.** Recall that, to show that the gradients of $J_1$ and $J_2$ (defined in Equation (15)) with respect to $w$ are equal, it suffices to show Equation (22). However, this is a direct consequence of Lemma 4. This is because we can rewrite Equation (22) as

$$\mathbb{E}_{\widetilde{D}\sim\mathcal{T}} \left[ \begin{array}{c} X^\top\vec{y}\hat{w}_{\widetilde{D}}A_{:,d+1} \\ \mathrm{tr}(\hat{w}_{\widetilde{D}}\vec{y}^\top X A^\top_{:,1:d}) \end{array} \right] = \mathbb{E}_{\widetilde{D}\sim\mathcal{T}} \left[ \begin{array}{c} X^\top\vec{y}\vec{y}^\top X A_{:,d+1} \\ \mathrm{tr}(X^\top\vec{y}\vec{y}^\top X A^\top_{:,1:d}) \end{array} \right]. \tag{26}$$

This shows that the gradients of $J_1$ and $J_2$ with respect to $w$ are equal, and we use similar arguments to show that the gradients of $J_1$ and $J_2$ with respect to $A$ are equal. As mentioned above, this implies that $\mathbb{E}_{\widetilde{D} \sim \mathcal{T}} \| M_{:,1:d}^\top G_{\widetilde{D}} w - \hat{w}_{\widetilde{D}} \|_2^2 - \mathbb{E}_{\widetilde{D} \sim \mathcal{T}} \| M_{:,1:d}^\top G_{\widetilde{D}} w - \eta X^\top \vec{y} \|_2^2$ is a constant that is independent of $M$ and $w$, as desired.

## 4 RESULTS FOR DIFFERENT DATA COVARIANCE MATRICES

In this section, we consider the setting where the $x_i$'s have a covariance that is different from the identity matrix, and we show that the loss is minimized when the one-layer transformer implements one step of gradient descent with preconditioning. This suggests that the distribution of the $x_i$'s has a significant effect on the algorithm that the transformer implements.

**Data Distribution.** Concretely, the data distribution is the same as before, but the $x_i$ are sampled from $\mathcal{N}(0, \Sigma)$, where $\Sigma \in \mathbb{R}^{d \times d}$ is a positive semi-definite (PSD) matrix. The outputs are generated according to $y_i = w^\top x_i + \epsilon_i$, where $w \sim \mathcal{N}(0, \Sigma^{-1})$. This can equivalently be written as $x_i = \Sigma^{1/2} u_i$, where $u_i \sim \mathcal{N}(0, I_{d \times d})$, and $y_i = (w')^\top u_i + \epsilon_i$, where $w' \sim \mathcal{N}(0, I_{d \times d})$. We keep all other definitions, such as the loss function, the same as before.

**Theorem 2** (Global Minimum for 1-Layer 1-Head Linear Self-Attention with Skewed Covariance). *Suppose $(W_K^*, W_Q^*, W_V^*, h^*)$ is a global minimizer of the loss $L$ when the data is generated according to the distribution given in this section. Then, the corresponding one-layer transformer implements one step of preconditioned gradient descent, on the least-squares linear regression objective, with preconditioner $\Sigma^{-1}$, for some learning rate $\eta > 0$. Specifically, given a query token $v_{n+1} = \begin{bmatrix} x_{n+1} \\ 0 \end{bmatrix}$, the transformer outputs $\eta \sum_{i=1}^n y_i (\Sigma^{-1} x_i)^\top x_{n+1}$, where $\eta = \frac{\mathbb{E}_{\widetilde{D} \sim \mathcal{T}} [\vec{y}^\top X (X^\top X + \sigma^2 \Sigma)^{-1} X^\top \vec{y}]}{\mathbb{E}_{\widetilde{D} \sim \mathcal{T}} [\vec{y}^\top X \Sigma^{-1} X^\top \vec{y}]}$.*

To prove this result, we essentially perform a change of variables to reduce this problem to the setting of the previous section — then, we directly apply Theorem 1. The detailed proof is given in Appendix D.

## 5 RESULTS FOR NONLINEAR TARGET FUNCTIONS

In this section, we extend to a setting where the target function is nonlinear — our conditions on the target function are mild, and for instance allow the target function to be a fully-connected neural network with arbitrary depth/width. However, we keep the model class the same (i.e. 1-layer transformer with linear self-attention). We find that the transformer which minimizes the pre-training loss still implements one step of GD on the *linear regression objective* (Theorem 3), even though the target function is nonlinear. This suggests that the distribution of $y_i | x_i$ does not affect the algorithm learned by the transformer as much as the distribution of $x_i$.

**Data Distribution.** In this section, we consider the same setup as Section 3, but we change the distribution of the $y_i$'s. We now assume $y_i = f(x_i) + \epsilon_i$, where $\epsilon_i \sim \mathcal{N}(0, \sigma^2)$ as before, but $f$ is drawn from a family of nonlinear functions satisfying the following assumption:

**Assumption 1.** *We assume that the target function $f$ is drawn from a family $\mathcal{F}$, with a probability measure $\mathbb{P}$ on $\mathcal{F}$, such that the following conditions hold: (1) for any fixed rotation matrix $R \in \mathbb{R}^{d \times d}$, the distribution of functions $f$ is the same as the distribution of $f \circ R$ (where $\circ$ denotes function composition). Moreover, the distribution of $f$ is symmetric under negation. In other words, if $E \subset \mathcal{F}$ is measurable under $\mathbb{P}$, then $\mathbb{P}(E) = \mathbb{P}(-E)$, where $-E = \{-f \mid f \in E\}$.*

For example, Assumption 1 is satisfied when $f(x)$ is a fully connected neural network, with arbitrary depth and width, where the first and last layers have i.i.d. $\mathcal{N}(0, 1)$ entries — see Appendix B for further discussion. Under this assumption, we prove the following result:

**Theorem 3** (Global Minimum for 1-Layer 1-Head Linear Self-Attention with Nonlinear Target Function). *Suppose Assumption 1 holds, and let $(W_K^*, W_Q^*, W_V^*, h^*)$ be a global minimizer of the pre-training loss. Then, the corresponding one-layer transformer implements one step of gradient descent on the least-squares linear regression objective, given $(x_1, y_1, \ldots, x_n, y_n)$. More con-*

*cretely, given a query token* $v_{n+1} = \begin{bmatrix} x_{n+1} \\ 0 \end{bmatrix}$, *the transformer outputs* $\eta \sum_{i=1}^{n} y_i x_i^\top x_{n+1}$, *where*

$\eta = \frac{\mathbb{E}_{\mathcal{D}}[\overline{u}_{\widetilde{D}}^\top X^\top \vec{y}]}{\mathbb{E}_{\mathcal{D}}[\vec{y}^\top X X^\top \vec{y}]}$, $\overline{u}_{\widetilde{D}} = \operatorname{argmin}_u \mathbb{E}_{x_{n+1},y_{n+1}}[(u \cdot x_{n+1} - y_{n+1})^2 \mid \widetilde{D}]$, *and* $\mathcal{D}, \widetilde{D}$ *are as in Section 2.*

The result is essentially the same as that of Theorem 1 — note that the learning rate is potentially different, as it may depend on the function family $\mathcal{F}$. The proof is analogous to the proof of Theorem 1. First we prove the analogue of Lemma 1, defining $L(w, M)$ as in Section 2:

**Lemma 5.** *There exists a constant* $C \geq 0$ *such that* $L(w, M) = C + \mathbb{E}_{\widetilde{D} \sim \mathcal{T}} \|M_{:,1:d}^\top G_{\widetilde{D}} w - \overline{u}_{\widetilde{D}}\|_2^2$,
*where* $\overline{u}_{\widetilde{D}} = \operatorname{argmin}_u \mathbb{E}_{x_{n+1},y_{n+1}}[(u \cdot x_{n+1} - y_{n+1})^2 \mid \widetilde{D}]$.

Next, in the proof of Lemma 2, we used the fact that odd-degree polynomials of the $y_i$ have expectation 0 — the corresponding lemma in our current setting is as follows:

**Lemma 6.** *For even integers* $k$ *and for* $i \in [n]$, $\mathbb{E}[y_i^k \overline{u}_{\widetilde{D}} \mid X] = 0$. *This also holds with* $y_i^k$ *replaced by an even-degree monomial of the* $y_i$. *Additionally, for odd integers* $k$ *and for* $i \in [n]$, $\mathbb{E}[y_i^k \mid X] = 0$. *This also holds with* $y_i^k$ *replaced by an odd-degree monomial of the* $y_i$.

*Proof of Lemma 6.* This follows from Assumption 1. This is because for each outcome (i.e. choice of $f$ and $\epsilon_1, \ldots, \epsilon_n$) which leads to $(x_1, y_1, \ldots, x_n, y_n)$, the corresponding outcome $-f, -\epsilon_1, \ldots, -\epsilon_n$ which leads to $(x_1, -y_1, \ldots, x_n, -y_n)$ is equally likely. The $\overline{u}_{\widetilde{D}}$ which is obtained from the second outcome is the negative of the $\overline{u}_{\widetilde{D}}$ which is obtained from the first outcome. If $k$ is even, then $y_i^k$ is the same under both outcomes since $k$ is even, and the average of $y_i^k \overline{u}_{\widetilde{D}}$ under these two outcomes is 0. Additionally, if $k$ is odd, then $y_i^k$ under the second outcome is the negative of $y_i^k$ under the first outcome, and the average of $y_i^k$ under these two outcomes is 0. This completes the proof of the lemma. $\square$

Next, we show the analogue of Lemma 3.
**Lemma 7.** $\mathbb{E}_{\widetilde{D} \sim \mathcal{T}}[X^\top \vec{y} \vec{y}^\top X]$ *and* $\mathbb{E}_{\widetilde{D} \sim \mathcal{T}}[X^\top \vec{y} \overline{u}_{\widetilde{D}}^\top]$ *are scalar multiples of the identity. Thus,*
$\mathbb{E}_{\widetilde{D} \sim \mathcal{T}}[X^\top \vec{y} \overline{u}_{\widetilde{D}}^\top] = \eta \mathbb{E}_{\widetilde{D} \sim \mathcal{T}}[X^\top \vec{y} \vec{y}^\top X]$, *where* $\eta = \frac{\mathbb{E}_{\mathcal{D}}[\overline{u}_{\widetilde{D}}^\top X^\top \vec{y}]}{\mathbb{E}_{\mathcal{D}}[\vec{y}^\top X X^\top \vec{y}]}$.

The proof of Lemma 7 is nearly identical to that of Lemma 3, with Assumption 1 used where appropriate. We include the proof in Appendix E. We now state the analogue of Lemma 2:

**Lemma 8.** *There exists a constant* $C_1 \geq 0$ *which is independent of* $w, M$, *such that* $L(w, M) = C + \mathbb{E}_{\widetilde{D} \sim \mathcal{T}} \|M_{:,1:d}^\top G_{\widetilde{D}} w - \eta X^\top \vec{y}\|_2^2$, *where* $\eta = \frac{\mathbb{E}_{\mathcal{D}}[\overline{u}_{\widetilde{D}}^\top X^\top \vec{y}]}{\mathbb{E}_{\mathcal{D}}[\vec{y}^\top X X^\top \vec{y}]}$.

Theorem 3 now follows from Lemma 8 because $M_{:,1:d}^\top G_{\widetilde{D}} w$ is the weight vector for the effective linear predictor implemented by the transformer. All missing proofs are in Appendix E.

## 6 CONCLUSION

We theoretically study one-layer transformers with linear self-attention trained on noisy linear regression tasks with randomly generated data. We confirm the empirical finding of von Oswald et al. (2022) by mathematically showing that the global minimum of the pre-training loss for one-layer transformers with linear self-attention corresponds to one step of GD on a least-squares linear regression objective, when the covariates are from an isotropic Gaussian distribution. We find that when the covariates are not from an isotropic Gaussian distribution, the global minimum of the pre-training loss instead corresponds to pre-conditioned GD, while if the covariates are from an isotropic Gaussian distribution and the response is obtained from a *nonlinear* target function, then the global minimum of the pre-training loss will still correspond to one step of GD on a least-squares linear regression objective. We study single-head linear self-attention layers — it is an interesting direction for future work to study the global minima of the pre-training loss for a multi-head linear self-attention layer. Another interesting direction is to study the algorithms learned by multi-layer transformers when the response is obtained from a *nonlinear* target function. We note that Ahn et al. (2023) have studied the case of multi-layer transformers when the target function is linear. They show that for certain restricted parameterizations of multi-layer linear transformers, the global minima or critical points of the pre-training loss correspond to interpretable gradient-based algorithms.

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

## A  ADDITIONAL RELATED WORK

The work of Takakura & Suzuki (2023) theoretically shows that transformers can achieve low approximation error when the target function is shift-equivariant on sequences of infinite length, subject to certain regularity conditions. Bai et al. (2023) propose various constructions through which transformers can solve problems such as Bayesian linear regression and generalized linear models, achieving low test error - they also show that transformers can represent a form of model selection. Guo et al. (2023) similarly provide constructions of transformers which can represent certain algorithms. Compared to these works which give approximation-theoretic constructions, we show that one step of gradient descent is the *best* predictor that can be implemented by 1 layer of linear self-attention.

Another concurrent work by Tarzanagh et al. (2023) analyzes the global minima and training dynamics of transformer layers with general data. They show that the global minimum corresponds with a type of max-margin solution, and give conditions under which the training dynamics converge to this global minimum.

## B  DISCUSSION OF ASSUMPTION 1

Assumption 1 holds in the case where $f$ is a fully connected neural network, where the first and last layers have i.i.d. $\mathcal{N}(0,1)$ entries. For simplicity, we show this in the case of two-layer neural networks. We let our function family $\mathcal{F}$ be $\{x \mapsto W_2\sigma(W_1x) \mid W_1, W_2 \text{ have i.i.d. } \mathcal{N}(0,1) \text{ entries}\}$. Let $f(x) = W_2\sigma(W_1x)$ be a two-layer neural network drawn randomly from $\mathcal{F}$. Let $R$ be a fixed rotation matrix. Then, $(f \circ R)(x) = W_2\sigma(W_1Rx)$. Since $R$ is a fixed rotation matrix, and $W_1$ has i.i.d. $\mathcal{N}(0,1)$ entries, $W_1R$ has the same distribution as $W_1$. Thus, $f \circ R$ has the same distribution as $f$.

## C  MISSING PROOFS FROM SECTION 3

*Proof of Lemma 1.* We can write

$$L(w, M) = \mathbb{E}_{D \sim \mathcal{T}}[(y_{n+1} - w^\top G_{\widetilde{D}} M v_{n+1})^2] \tag{27}$$

$$= \mathbb{E}_{\widetilde{D}, x_{n+1}}\Big[\mathbb{E}_{y_{n+1}}[(y_{n+1} - w^\top G_{\widetilde{D}} M v_{n+1})^2 \mid x_{n+1}, \widetilde{D}]\Big]. \tag{28}$$

Let us simplify the inner expectation. For convenience, fix $\widetilde{D}$ and $x_{n+1}$, and consider the function $g : \mathbb{R}^d \to \mathbb{R}$ given by

$$g(u) = \mathbb{E}_{y_{n+1}}[(u \cdot x_{n+1} - y_{n+1})^2 \mid \widetilde{D}, x_{n+1}]. \tag{29}$$

It is a well-known fact that the minimizer of $g(u)$ is given by $\hat{w}_{\widetilde{D}}$ where $\hat{w}_{\widetilde{D}} = (X^\top X + \sigma^2 I)^{-1} X^\top \vec{y}$ is the solution to ridge regression on $X$ and $\vec{y}$ with regularization strength $\sigma^2$. (See e.g. equation (19) of Akyürek et al. (2023).) Furthermore,

$$0 = \nabla_u g(\hat{w}_{\widetilde{D}}) = \mathbb{E}_{y_{n+1}}[2(\hat{w}_{\widetilde{D}} \cdot x_{n+1} - y_{n+1})x_{n+1} \mid \widetilde{D}, x_{n+1}], \tag{30}$$

and in particular, taking the dot product of both sides with $u - \hat{w}_{\widetilde{D}}$ (for any vector $u \in \mathbb{R}^d$) gives

$$\mathbb{E}_{y_{n+1}}[(\hat{w}_{\widetilde{D}} \cdot x_{n+1} - y_{n+1}) \cdot (u \cdot x_{n+1} - \hat{w}_{\widetilde{D}} \cdot x_{n+1})] = 0 \,. \tag{31}$$

Thus, letting $u = w^\top G_{\widetilde{D}} M_{:,1:d}$, we can simplify the inner expectation in Equation (28):

$$\mathbb{E}_{y_{n+1}}[(y_{n+1} - w^\top G_{\widetilde{D}} M_{:,1:d} x_{n+1})^2 \mid x_{n+1}, \widetilde{D}] \tag{32}$$

$$= \mathbb{E}_{y_{n+1}}[(y_{n+1} - \hat{w}_{\widetilde{D}}^\top x_{n+1} + \hat{w}_{\widetilde{D}}^\top x_{n+1} - w^\top G_{\widetilde{D}} M_{:,1:d} x_{n+1})^2 \mid x_{n+1}, \widetilde{D}] \tag{33}$$

$$= \mathbb{E}_{y_{n+1}}[(y_{n+1} - \hat{w}_{\widetilde{D}}^\top x_{n+1})^2 \mid x_{n+1}, \widetilde{D}] + (\hat{w}_{\widetilde{D}}^\top x_{n+1} - w^\top G_{\widetilde{D}} M_{:,1:d} x_{n+1})^2 \tag{34}$$

$$+ 2 \cdot \mathbb{E}_{y_{n+1}}[(y_{n+1} - \hat{w}_{\widetilde{D}}^\top x_{n+1})(\hat{w}_{\widetilde{D}}^\top x_{n+1} - w^\top G_{\widetilde{D}} M_{:,1:d} x_{n+1}) \mid x_{n+1}, \widetilde{D}] \tag{35}$$

$$= \mathbb{E}_{y_{n+1}}[(y_{n+1} - \hat{w}_{\widetilde{D}}^\top x_{n+1})^2 \mid x_{n+1}, \widetilde{D}] + (\hat{w}_{\widetilde{D}}^\top x_{n+1} - w^\top G_{\widetilde{D}} M_{:,1:d} x_{n+1})^2 \,.$$
$$\text{(By Equation (31))}$$

We can further write the final expression as $C_{x_{n+1}, \widetilde{D}} + (\hat{w}_{\widetilde{D}}^\top x_{n+1} - w^\top G_{\widetilde{D}} M_{:,1:d} x_{n+1})^2$, where $C_{x_{n+1}, \widetilde{D}}$ is a constant that depends on $x_{n+1}$ and $\widetilde{D}$ but is independent of $w$ and $M$. Thus, we have

$$L(w, M) = \mathbb{E}_{\widetilde{D}, x_{n+1}}[C_{x_{n+1}, \widetilde{D}}] + \mathbb{E}_{\widetilde{D}, x_{n+1}}[(\hat{w}_{\widetilde{D}}^\top x_{n+1} - w^\top G_{\widetilde{D}} M_{:,1:d} x_{n+1})^2] \tag{36}$$

$$= C + \mathbb{E}_{\widetilde{D}}\|\hat{w}_{\widetilde{D}} - M_{:,1:d}^\top G_{\widetilde{D}} w\|_2^2 \,, \qquad \text{(B.c. } x_{n+1} \sim \mathcal{N}(0, I_{d \times d}))$$

where $C$ is a constant which is independent of $w$ and $M$. $\qquad\square$

*Proof of Lemma 3.* For convenience let $M(X) = \mathbb{E}_{\widetilde{D} \sim \mathcal{T}}[\vec{y}\vec{y}^\top \mid X]$ — we use this notation to make the dependence on $X$ clear. Then, the $(i, j)$-th entry of $M(X)$ is $\mathbb{E}[(w \cdot x_i + \epsilon_i)(w \cdot x_j + \epsilon_j) \mid X]$ and this is equal to $\mathbb{E}[(w \cdot x_i)(w \cdot x_j) \mid X]$ for $i \neq j$, and $\mathbb{E}[(w \cdot x_i)(w \cdot x_j) \mid X] + \sigma^2$ for $i = j$. If we perform the change of variables $x_i \mapsto Rx_i$ for a fixed rotation matrix $R$ and all $i \in [n]$, then $\mathbb{E}_w[(w \cdot Rx_i)(w \cdot Rx_j) \mid X] = \mathbb{E}_w[(w \cdot x_i)(w \cdot x_j) \mid X]$ because $w \sim \mathcal{N}(0, I_{d \times d})$ and $\mathcal{N}(0, I_{d \times d})$ is a rotationally invariant distribution. In other words, we have $M(XR^\top) = M(X)$. Thus, for any rotation matrix $R$,

$$\mathbb{E}_X[X^\top M(X) X] = \mathbb{E}_X[(XR)^\top M(XR^\top)(XR^\top)] \qquad \text{(By rotational invariance of dist. of } x_i)$$

$$= R^\top \mathbb{E}_X[X^\top M(XR^\top) X] R^\top \tag{37}$$

$$= R^\top \mathbb{E}_X[X^\top M(X) X] R \,. \qquad \text{(Because } M(XR^\top) = M(X))$$

This implies that $\mathbb{E}_X[X^\top M(X) X] = \mathbb{E}_{\widetilde{D} \sim \mathcal{T}}[X^\top \vec{y}\vec{y}^\top X]$ is a scalar multiple of the identity matrix.

Next, we consider $\mathbb{E}_{\widetilde{D} \sim \mathcal{T}}[\vec{y}\hat{w}_{\widetilde{D}}^\top \mid X]$, which we write for convenience as $J(X)$. Similarly to the above proof, we use the observation that if we make the change of variables $x_i \to Rx_i$, then the joint distribution of $y_1, \ldots, y_n, y_{n+1}$ is unchanged due to the rotational invariance of $w$. Additionally, if we write $\hat{w}_{\widetilde{D}}$ as $\hat{w}_{\widetilde{D}}(x_1, y_1, \ldots, x_n, y_n)$ to emphasize the fact that it is a function of $x_1, y_1, \ldots, x_n, y_n$, then

$$\hat{w}_{\widetilde{D}}(Rx_1, y_1, Rx_2, y_2, \ldots, Rx_n, y_n) = R\hat{w}_{\widetilde{D}}(x_1, y_1, x_2, y_2, \ldots, x_n, y_n) \tag{38}$$

because $\hat{w}_{\widetilde{D}}$ is the minimizer of $F(w) = \|w^\top X - \vec{y}\|_2^2 + \sigma^2 \|w\|_2^2$, and if all the $x_i$ are rotated by $R$, then the minimizer of $F$ will also be rotated by $R$. Thus,

$$J(XR^\top) = \mathbb{E}_{\widetilde{D} \sim \mathcal{T}}[\vec{y}\hat{w}_{\widetilde{D}}^\top \mid XR^\top] = \mathbb{E}_{\widetilde{D} \sim \mathcal{T}}[\vec{y}\hat{w}_{\widetilde{D}}^\top R^\top \mid X] = J(X)R^\top \,. \tag{39}$$

We can use this to show that $\mathbb{E}_{\widetilde{D} \sim \mathcal{T}}[X^\top \vec{y}\hat{w}_{\widetilde{D}}^\top]$ is a scalar multiple of the identity. Letting $R$ be a rotation matrix, we obtain

$$\mathbb{E}_{\widetilde{D} \sim \mathcal{T}}[X^\top \vec{y}\hat{w}_{\widetilde{D}}^\top] = \mathbb{E}_{\widetilde{D} \sim \mathcal{T}}\mathbb{E}[X^\top \vec{y}\hat{w}_{\widetilde{D}}^\top \mid X] \tag{40}$$

$$= \mathbb{E}_{\widetilde{D} \sim \mathcal{T}}[X^\top \mathbb{E}[\vec{y}\hat{w}_{\widetilde{D}}^\top \mid X]] \tag{41}$$

$$= \mathbb{E}_X[X^\top J(X)] \tag{42}$$

$$= \mathbb{E}_X[(XR^\top)^\top J(XR^\top)] \qquad \text{(By rotational invariance of dist. of } x_i)$$

$$= \mathbb{E}_X[RX^\top J(X) R^\top] \qquad \text{(B.c. } J(XR^\top) = J(X)R^\top)$$

$$= R\mathbb{E}_X[X^\top J(X)] R^\top \tag{43}$$

$$= R\mathbb{E}_{\widetilde{D} \sim \mathcal{T}}[X^\top \vec{y}\hat{w}_{\widetilde{D}}^\top] R^\top \,. \qquad \text{(By Equation (42))}$$

This implies that $\mathbb{E}_{\widetilde{D} \sim \mathcal{T}}[X^\top \vec{y}\hat{w}_{\widetilde{D}}^\top]$ is a scalar multiple of the identity. $\qquad\square$

*Proof of Lemma 4.* By Lemma 3, we have $\mathbb{E}_{\widetilde{D}\sim\mathcal{T}}[X^\top \vec{y}\vec{y}^\top X] = c_1 I$ and $\mathbb{E}_{\widetilde{D}\sim\mathcal{T}}[X^\top \vec{y}\hat{w}_{\widetilde{D}}^\top] = c_2 I$ for some scalars $c_1, c_2$. Taking the traces of both matrices gives us

$$c_1 d = \text{tr}(\mathbb{E}_{\widetilde{D}\sim\mathcal{T}}[X^\top \vec{y}\vec{y}^\top X]) = \mathbb{E}_{\widetilde{D}\sim\mathcal{T}}[\text{tr}(X^\top \vec{y}\vec{y}^\top X)] = \mathbb{E}_{\widetilde{D}\sim\mathcal{T}}[\vec{y}^\top XX^\top \vec{y}], \tag{44}$$

and similarly,

$$c_2 d = \text{tr}(\mathbb{E}_{\widetilde{D}\sim\mathcal{T}}[X^\top \vec{y}\hat{w}_{\widetilde{D}}^\top]) = \mathbb{E}_{\widetilde{D}\sim\mathcal{T}}[\text{tr}(X^\top \vec{y}\hat{w}_{\widetilde{D}}^\top)] = \mathbb{E}_{\widetilde{D}\sim\mathcal{T}}[\hat{w}_{\widetilde{D}}^\top X^\top \vec{y}]. \tag{45}$$

By the definition of $\eta$ as $\frac{\mathbb{E}_{\widetilde{D}\sim\mathcal{T}}[\hat{w}_{\widetilde{D}}^\top X^\top \vec{y}]}{\mathbb{E}_{\widetilde{D}\sim\mathcal{T}}[\vec{y}^\top XX^\top \vec{y}]}$, we have

$$\eta\mathbb{E}_{\widetilde{D}\sim\mathcal{T}}[X^\top \vec{y}\vec{y}^\top X] - \mathbb{E}_{\widetilde{D}\sim\mathcal{T}}[X^\top \vec{y}\hat{w}_{\widetilde{D}}^\top] = \eta c_1 I - c_2 I \tag{46}$$

$$= \eta \cdot \frac{\mathbb{E}_{\widetilde{D}\sim\mathcal{T}}[\vec{y}^\top XX^\top \vec{y}]}{d} I - \frac{\mathbb{E}_{\widetilde{D}\sim\mathcal{T}}[\hat{w}_{\widetilde{D}}^\top X^\top \vec{y}]}{d} I \tag{47}$$

$$= \frac{\mathbb{E}_{\widetilde{D}\sim\mathcal{T}}[\hat{w}_{\widetilde{D}}^\top X^\top \vec{y}]}{d} I - \frac{\mathbb{E}_{\widetilde{D}\sim\mathcal{T}}[\hat{w}_{\widetilde{D}}^\top X^\top \vec{y}]}{d} I \tag{48}$$

$$= 0, \tag{49}$$

completing the proof of the lemma. $\square$

*Proof of Lemma 2.* For convenience, we write $A := M_{:,1:d}^\top$. Then, we wish to show that for all $A \in \mathbb{R}^{d\times(d+1)}$ and $w \in \mathbb{R}^{d+1}$, we have

$$\mathbb{E}_{\widetilde{D}\sim\mathcal{T}}\|AG_{\widetilde{D}}w - \hat{w}_{\widetilde{D}}\|_2^2 = C + \mathbb{E}_{\widetilde{D}\sim\mathcal{T}}\|AG_{\widetilde{D}}w - \eta X^\top \vec{y}\|_2^2 \tag{50}$$

for some constant $C$ independent of $A$ and $w$. Define

$$J_1(A, w) = \mathbb{E}_{\widetilde{D}\sim\mathcal{T}}\|AG_{\widetilde{D}}w - \hat{w}_{\widetilde{D}}\|_2^2 \tag{51}$$

and

$$J_2(A, w) = \mathbb{E}_{\widetilde{D}\sim\mathcal{T}}\|AG_{\widetilde{D}}w - \eta X^\top \vec{y}\|_2^2. \tag{52}$$

To show that $J_1(A, w)$ and $J_2(A, w)$ are equal up to a constant, it suffices to show that their gradients are identical, since $J_1$ and $J_2$ are defined everywhere on $\mathbb{R}^{d+1} \times \mathbb{R}^{(d+1)\times(d+1)}$ and by the fundamental theorem of calculus.

**Gradients With Respect to $w$.** First, we analyze the gradients with respect to $w$. We have

$$\nabla_w J_1(A, w) = 2\mathbb{E}_{\widetilde{D}\sim\mathcal{T}}G_{\widetilde{D}}A^\top(AG_{\widetilde{D}}w - \hat{w}_{\widetilde{D}}) \tag{53}$$

and

$$\nabla_w J_2(A, w) = 2\mathbb{E}_{\widetilde{D}\sim\mathcal{T}}G_{\widetilde{D}}A^\top(AG_{\widetilde{D}}w - \eta X^\top \vec{y}) \tag{54}$$

where we use the convention that the gradient with respect to $w$ has the same shape as $w$. To show that $\nabla_w J_1(A, w) = \nabla_w J_2(A, w)$, it suffices to show that $\mathbb{E}_{\widetilde{D}\sim\mathcal{T}}G_{\widetilde{D}}A^\top\hat{w}_{\widetilde{D}} = \eta\mathbb{E}_{\widetilde{D}\sim\mathcal{T}}G_{\widetilde{D}}A^\top X^\top \vec{y}$. Observe that we can write $G_{\widetilde{D}}$ as

$$G_{\widetilde{D}} = \sum_{i=1}^n \begin{bmatrix} x_i \\ y_i \end{bmatrix} \begin{bmatrix} x_i \\ y_i \end{bmatrix}^\top = \sum_{i=1}^n \begin{bmatrix} x_i x_i^\top & y_i x_i \\ y_i x_i^\top & y_i^2 \end{bmatrix}. \tag{55}$$

Given $X$, the expected value of any odd monomial of the $y_i$ is equal to 0, since $w$ and $\epsilon_i$ are independent of $X$ and have mean 0. Thus, we can ignore the blocks corresponding to $x_i x_i^\top$ and $y_i^2$, since the entries of $\hat{w}_{\widetilde{D}}$ and $X^\top \vec{y}$ contain odd monomials of the $y_i$ once $X$ is fixed.

We now proceed in two steps. First, we deal with the terms corresponding to the lower left block of $G_{\widetilde{D}}$ and show that their respective contributions to $\mathbb{E}_{\widetilde{D}\sim\mathcal{T}}G_{\widetilde{D}}A^\top\hat{w}_{\widetilde{D}}$ and $\eta\mathbb{E}_{\widetilde{D}\sim\mathcal{T}}G_{\widetilde{D}}A^\top X^\top \vec{y}$ are equal. For $\mathbb{E}_{\widetilde{D}\sim\mathcal{T}}G_{\widetilde{D}}A^\top\hat{w}_{\widetilde{D}}$, these terms contribute:

$$\mathbb{E}_{\widetilde{D}\sim\mathcal{T}}\sum_{i=1}^n y_i x_i^\top(A^\top)_{1:d,:}\hat{w}_{\widetilde{D}} = \mathbb{E}_{\widetilde{D}\sim\mathcal{T}}\vec{y}^\top XA_{:,1:d}^\top\hat{w}_{\widetilde{D}} \tag{56}$$

$$= \mathbb{E}_{\widetilde{D}\sim\mathcal{T}}\text{tr}(\hat{w}_{\widetilde{D}}\vec{y}^\top XA_{:,1:d}^\top) \tag{57}$$

$$= \text{tr}(\mathbb{E}_{\widetilde{D}\sim\mathcal{T}}[\hat{w}_{\widetilde{D}}\vec{y}^\top X]A_{:,1:d}^\top), \tag{58}$$

while for $\eta\mathbb{E}_{\widetilde{D}\sim\mathcal{T}}G_{\widetilde{D}}A^\top X^\top\vec{y}$, these terms contribute:

$$\eta\mathbb{E}_{\widetilde{D}\sim\mathcal{T}}\sum_{i=1}^n y_i x_i^\top A_{:,1:d}^\top X^\top\vec{y} = \eta\mathbb{E}_{\widetilde{D}\sim\mathcal{T}}\vec{y}^\top X A_{:,1:d}^\top X^\top\vec{y} \tag{59}$$

$$= \eta\mathbb{E}_{\widetilde{D}\sim\mathcal{T}}\text{tr}(X^\top\vec{y}\vec{y}^\top X A_{:,1:d}^\top) \tag{60}$$

$$= \text{tr}(\mathbb{E}_{\widetilde{D}\sim\mathcal{T}}[\eta X^\top\vec{y}\vec{y}^\top X]A_{:,1:d}^\top), \tag{61}$$

and since $\mathbb{E}_{\widetilde{D}\sim\mathcal{T}}[\hat{w}_{\widetilde{D}}\vec{y}^\top X] = \mathbb{E}_{\widetilde{D}\sim\mathcal{T}}[\eta X^\top\vec{y}\vec{y}^\top X]$ by Lemma 4, these two contributions are equal.

Second, we deal with the terms corresponding to the upper right block of $G_{\widetilde{D}}$. For $\mathbb{E}_{\widetilde{D}\sim\mathcal{T}}G_{\widetilde{D}}A^\top\hat{w}_{\widetilde{D}}$ these terms contribute

$$\mathbb{E}_{\widetilde{D}\sim\mathcal{T}}\sum_{i=1}^n y_i x_i(A^\top)_{(d+1),:}\hat{w}_{\widetilde{D}} = \mathbb{E}_{\widetilde{D}\sim\mathcal{T}}X^\top\vec{y}A_{:,(d+1)}^\top\hat{w}_{\widetilde{D}} \tag{62}$$

$$= \mathbb{E}_{\widetilde{D}\sim\mathcal{T}}X^\top\vec{y}\hat{w}_{\widetilde{D}}^\top A_{:,(d+1)},$$
$$\text{(B.c. dot product } A_{:,(d+1)}^\top\hat{w}_{\widetilde{D}} \text{ is commutative)}$$

while for $\eta\mathbb{E}_{\widetilde{D}\sim\mathcal{T}}G_{\widetilde{D}}A^\top X^\top\vec{y}$, these terms contribute

$$\eta\mathbb{E}_{\widetilde{D}\sim\mathcal{T}}\sum_{i=1}^n y_i x_i(A^\top)_{(d+1),:}X^\top\vec{y} = \eta\mathbb{E}_{\widetilde{D}\sim\mathcal{T}}X^\top\vec{y}A_{:,(d+1)}^\top X^\top\vec{y} \tag{63}$$

$$= \eta\mathbb{E}_{\widetilde{D}\sim\mathcal{T}}X^\top\vec{y}\vec{y}^\top X A_{:,(d+1)},$$
$$\text{(B.c. dot product } A_{:,(d+1)}^\top X^\top\vec{y} \text{ is commutative)}$$

and again these contributions are equal by Lemma 4. In summary, we have shown that for $\mathbb{E}_{\widetilde{D}\sim\mathcal{T}}G_{\widetilde{D}}A^\top\hat{w}_{\widetilde{D}}$ and $\eta\mathbb{E}_{\widetilde{D}\sim\mathcal{T}}G_{\widetilde{D}}A^\top X^\top\vec{y}$, the contributions corresponding to the lower left and upper right blocks of $G_{\widetilde{D}}$ are equal, meaning that $\mathbb{E}_{\widetilde{D}\sim\mathcal{T}}G_{\widetilde{D}}A^\top\hat{w}_{\widetilde{D}}$ and $\eta\mathbb{E}_{\widetilde{D}\sim\mathcal{T}}G_{\widetilde{D}}A^\top X^\top\vec{y}$ are equal. This shows that $\nabla_w J_1(A,w)$ and $\nabla_w J_2(A,w)$ are equal.

**Gradients With Respect to $A$.** We can compute the gradient of $J_1$ with respect to $A$ as follows:

$$\nabla_A J_1(A,w) = \mathbb{E}_{\widetilde{D}\sim\mathcal{T}}\nabla_A\|AG_{\widetilde{D}}w - \hat{w}_{\widetilde{D}}\|_2^2 \tag{64}$$

$$= \mathbb{E}_{\widetilde{D}\sim\mathcal{T}}\nabla_A(AG_{\widetilde{D}}w - \hat{w}_{\widetilde{D}})^\top(AG_{\widetilde{D}}w - \hat{w}_{\widetilde{D}}) \tag{65}$$

$$= 2\mathbb{E}_{\widetilde{D}\sim\mathcal{T}}(AG_{\widetilde{D}}w - \hat{w}_{\widetilde{D}})(G_{\widetilde{D}}w)^\top, \tag{66}$$

and we can similarly compute

$$\nabla_A J_2(A,w) = \mathbb{E}_{\widetilde{D}\sim\mathcal{T}}\nabla_A\|AG_{\widetilde{D}}w - \eta X^\top\vec{y}\|_2^2 \tag{67}$$

$$= \mathbb{E}_{\widetilde{D}\sim\mathcal{T}}\nabla_A(AG_{\widetilde{D}}w - \eta X^\top\vec{y})^\top(AG_{\widetilde{D}}w - \eta X^\top\vec{y}) \tag{68}$$

$$= 2\mathbb{E}_{\widetilde{D}\sim\mathcal{T}}(AG_{\widetilde{D}}w - \eta X^\top\vec{y})(G_{\widetilde{D}}w)^\top. \tag{69}$$

Thus, to show that the gradients with respect to $A$ are equal, it suffices to show that

$$\mathbb{E}_{\widetilde{D}\sim\mathcal{T}}\hat{w}_{\widetilde{D}}w^\top G_{\widetilde{D}} = \eta\mathbb{E}_{\widetilde{D}\sim\mathcal{T}}X^\top\vec{y}w^\top G_{\widetilde{D}} \tag{70}$$

for all $w$.

As before, we only need to consider the lower left and upper right blocks of $G_{\widetilde{D}}$, since the entries of the other blocks have even powers of the $y_i$. First let us consider the lower left block. The contribution of the lower left block to $\mathbb{E}_{\widetilde{D}\sim\mathcal{T}}\hat{w}_{\widetilde{D}}w^\top G_{\widetilde{D}}$ is

$$\mathbb{E}_{\widetilde{D}\sim\mathcal{T}}\hat{w}_{\widetilde{D}}w_{(d+1)}\sum_{i=1}^n y_i x_i^\top = \mathbb{E}_{\widetilde{D}\sim\mathcal{T}}\hat{w}_{\widetilde{D}}w_{(d+1)}\vec{y}^\top X = w_{(d+1)}\mathbb{E}_{\widetilde{D}\sim\mathcal{T}}\hat{w}_{\widetilde{D}}\vec{y}^\top X, \tag{71}$$

while the contribution of the lower left block to $\eta\mathbb{E}_{\widetilde{D}\sim\mathcal{T}}X^\top\vec{y}w^\top G_{\widetilde{D}}$ is

$$\mathbb{E}_{\widetilde{D}\sim\mathcal{T}}\eta X^\top\vec{y}w_{(d+1)}\sum_{i=1}^n y_i x_i^\top = \eta w_{(d+1)}\mathbb{E}_{\widetilde{D}\sim\mathcal{T}}X^\top\vec{y}\vec{y}^\top X, \tag{72}$$

and these two are equal by Lemma 4. The contribution of the upper right block to $\mathbb{E}_{\widetilde{D}\sim\mathcal{T}}\hat{w}_{\widetilde{D}}w^\top G_{\widetilde{D}}$ is

$$\mathbb{E}_{\widetilde{D}\sim\mathcal{T}}\hat{w}_{\widetilde{D}}w_{1:d}^\top \sum_{i=1}^n y_i x_i = \mathbb{E}_{\widetilde{D}\sim\mathcal{T}}\hat{w}_{\widetilde{D}}w_{1:d}^\top X^\top \vec{y} = (\mathbb{E}_{\widetilde{D}\sim\mathcal{T}}\hat{w}_{\widetilde{D}}\vec{y}^\top X)w_{1:d}, \tag{73}$$

by the commutativity of the dot product $w_{1:d}^\top X^\top \vec{y}$, while similarly, the contribution of the upper right block to $\eta\mathbb{E}_{\widetilde{D}\sim\mathcal{T}}X^\top \vec{y}w^\top G_{\widetilde{D}}$ is

$$\mathbb{E}_{\widetilde{D}\sim\mathcal{T}}\eta X^\top \vec{y}w_{1:d}^\top \sum_{i=1}^n y_i x_i = (\eta\mathbb{E}_{\widetilde{D}\sim\mathcal{T}}X^\top \vec{y}\vec{y}^\top X)w_{1:d}, \tag{74}$$

and these two quantities are equal by Lemma 4. Thus, we have shown that $\nabla_A J_1(A,w) = \nabla_A J_2(A,w)$.

**Summary.** We have shown that $\nabla_w J_1(A,w) = \nabla_w J_2(A,w)$, and $\nabla_A J_1(A,w) = \nabla_A J_2(A,w)$. This completes the proof of the lemma. $\qquad\square$

*Proof of Theorem 1.* This theorem follows from Lemma 1 and Lemma 2. To see why, recall from Section 2 that the output of the transformer on the last token $v_{n+1}$ can be written as $w^\top G_{\widetilde{D}}Mv_{n+1}$, where $M = W_K^\top W_Q$ and $w = W_V^\top h$. By Lemma 2, the pre-training loss is minimized when $M_{:,1:d}^\top G_{\widetilde{D}}w = \eta X^\top \vec{y}$ almost surely (over the randomness of $X$ and $\vec{y}$), and when this holds, the output of the transformer on the last token is

$$w^\top G_{\widetilde{D}}Mv_{n+1} = w^\top G_{\widetilde{D}}M_{:,1:d}x_{n+1} = (\eta X^\top \vec{y})^\top x_{n+1} = \eta\sum_{i=1}^n y_i x_i^\top x_{n+1}, \tag{75}$$

as desired. $\qquad\square$

# D  MISSING PROOFS FOR SECTION 4

*Proof of Theorem 2.* As discussed in Section 4, we can write $x_i = \Sigma^{1/2}u_i$, where $u_i \sim \mathcal{N}(0, I_{d\times d})$. Furthermore, we can let $U \in \mathbb{R}^{n\times d}$ be the matrix whose $i^{\text{th}}$ row is $u_i^\top$. Note that $X = U\Sigma^{1/2}$, since $\Sigma$ is a symmetric positive-definite matrix. Given $\widetilde{D} = (x_1, y_1, \ldots, x_n, y_n)$, we can define $\hat{w}_{\widetilde{D},\Sigma} = (U^\top U + \sigma^2 I)^{-1}U^\top \vec{y}$ — this would be the solution to ridge regression if we were given the $u_i$ and $y_i$. Using $X = U\Sigma^{1/2}$ which implies $U = X\Sigma^{-1/2}$, we obtain

$$\hat{w}_{\widetilde{D},\Sigma} = (\Sigma^{-1/2}X^\top X\Sigma^{-1/2} + \sigma^2 I)^{-1}\Sigma^{-1/2}X^\top \vec{y} \tag{76}$$

$$= (\Sigma^{1/2}\Sigma^{-1/2}X^\top X\Sigma^{-1/2} + \sigma^2\Sigma^{1/2})^{-1}X^\top \vec{y} \tag{77}$$

$$= (X^\top X\Sigma^{-1/2} + \sigma^2\Sigma^{1/2})^{-1}X^\top \vec{y} \tag{78}$$

$$= \Sigma^{1/2}(X^\top X + \sigma^2\Sigma)^{-1}X^\top \vec{y}. \tag{79}$$

We now change variables in order to reduce Theorem 2 to Theorem 1, using the fact that $u_i$ and $y_i$ together are from the same distribution as the data studied in Theorem 1. We can write the loss as

$$L(W_K, W_Q, W_V, h) = \mathbb{E}_{D\sim\mathcal{T}}[(y_{n+1} - h^\top \hat{v}_{n+1})^2] \tag{80}$$

$$= \mathbb{E}_{\widetilde{D},x_{n+1}}\left[\mathbb{E}_{y_{n+1}}[(y_{n+1} - h^\top \hat{v}_{n+1})^2 \mid x_{n+1}, \mathcal{D}]\right]. \tag{81}$$

The solution to ridge regression given the $u_i$ and $y_i$ is $\hat{w}_{\widetilde{D},\Sigma}$, meaning $\mathbb{E}[y_{n+1} \mid u_{n+1}, \widetilde{D}] = \hat{w}_{\widetilde{D},\Sigma}^\top u_{n+1}$ and therefore $\mathbb{E}[y_{n+1} \mid x_{n+1}, \widetilde{D}] = \hat{w}_{\widetilde{D},\Sigma}^\top u_{n+1}$, since $u_{n+1} = \Sigma^{-1/2}x_{n+1}$, which is an invertible function of $x_{n+1}$. Thus,

$$\mathbb{E}_{y_{n+1}}[(y_{n+1} - h^\top \hat{v}_{n+1})^2 \mid x_{n+1}, \widetilde{D}] \tag{82}$$

$$= \mathbb{E}_{y_{n+1}}[(y_{n+1} - \hat{w}_{\widetilde{D},\Sigma}^\top u_{n+1})^2 \mid x_{n+1}, \widetilde{D}] \tag{83}$$

$$+ \mathbb{E}_{y_{n+1}}[(\hat{w}_{\widetilde{D},\Sigma}^\top u_{n+1} - h^\top \hat{v}_{n+1})^2 \mid x_{n+1}, \widetilde{D}] \tag{84}$$

$$+ 2\mathbb{E}_{y_{n+1}}[(y_{n+1} - \hat{w}_{\widetilde{D},\Sigma}^\top u_{n+1})(\hat{w}_{\widetilde{D},\Sigma}^\top u_{n+1} - h^\top v_{n+1}) \mid x_{n+1}, \widetilde{D}]. \tag{85}$$

Now, note that $\mathbb{E}_{y_{n+1}}[(y_{n+1} - \hat{w}_{\widetilde{D},\Sigma}^{\top} u_{n+1})(\hat{w}_{\widetilde{D},\Sigma}^{\top} u_{n+1} - h^{\top} v_{n+1}) \mid x_{n+1}, \widetilde{D}] = 0$ since $\hat{w}_{\widetilde{D},\Sigma}^{\top} u_{n+1} - h^{\top} \hat{v}_{n+1}$ is fully determined by $x_{n+1}$ and $\widetilde{D}$, and $\mathbb{E}[y_{n+1} - \hat{w}_{\widetilde{D},\Sigma}^{\top} u_{n+1} \mid x_{n+1}, \widetilde{D}] = 0$ as mentioned above. Additionally, we can write $\mathbb{E}_{y_{n+1}}[(y_{n+1} - \hat{w}_{\widetilde{D},\Sigma}^{\top} u_{n+1})^2 \mid x_{n+1}, \widetilde{D}]$ as $C_{x_{n+1},\widetilde{D}}$ (denoting a constant that depends only on $x_{n+1}$ and $\widetilde{D}$) since it is independent of the parameters $W_K, W_Q, W_V, h$. Thus, taking the expectation over $x_{n+1}$ and $\widetilde{D}$, for some constant $C$ which is independent of $W_K, W_Q, W_V, h$, we have

$$L(W_K, W_Q, W_V, h) = C + \mathbb{E}_{\widetilde{D}, x_{n+1}}(\hat{w}_{\widetilde{D},\Sigma}^{\top} u_{n+1} - h^{\top} \hat{v}_{n+1})^2 \tag{86}$$

$$= C + \mathbb{E}_{\widetilde{D}, x_{n+1}}(\hat{w}_{\widetilde{D},\Sigma}^{\top} u_{n+1} - h^{\top} W_V G_{\widetilde{D}} W_K^{\top} W_Q v_{n+1})^2 \tag{87}$$

$$= C + \mathbb{E}_{\widetilde{D}, x_{n+1}}(\hat{w}_{\widetilde{D},\Sigma}^{\top} u_{n+1} - h^{\top} W_V G_{\widetilde{D}} W_K^{\top} W_{Q_{:,1:d}} x_{n+1})^2 \tag{88}$$

$$= C + \mathbb{E}_{\widetilde{D}, x_{n+1}}(\hat{w}_{\widetilde{D},\Sigma}^{\top} u_{n+1} - h^{\top} W_V G_{\widetilde{D}} W_K^{\top} W_{Q_{:,1:d}} \Sigma^{1/2} u_{n+1})^2 . \tag{89}$$

To finish the proof, observe that we can write

$$G_{\widetilde{D}} = \sum_{i=1}^{n} \begin{bmatrix} x_i \\ y_i \end{bmatrix} \begin{bmatrix} x_i \\ y_i \end{bmatrix}^{\top} \tag{90}$$

$$= \begin{pmatrix} \Sigma^{1/2} & 0 \\ 0 & 1 \end{pmatrix} \sum_{i=1}^{n} \begin{bmatrix} u_i \\ y_i \end{bmatrix} \begin{bmatrix} u_i \\ y_i \end{bmatrix}^{\top} \begin{pmatrix} \Sigma^{1/2} & 0 \\ 0 & 1 \end{pmatrix} \tag{91}$$

$$= H G_{\widetilde{D}}' H^{\top} , \tag{92}$$

where we have defined $H = \begin{pmatrix} \Sigma^{1/2} & 0 \\ 0 & 1 \end{pmatrix}$ and

$$G_{\widetilde{D}}' = \sum_{i=1}^{n} \begin{bmatrix} u_i \\ y_i \end{bmatrix} \begin{bmatrix} u_i \\ y_i \end{bmatrix}^{\top} . \tag{93}$$

Now, let us make the change of variables $h = h'$, $W_V = W_V' H^{-1}$, $W_K = W_K' H^{-1}$ and $W_Q = W_Q' H^{-1}$. Then, we obtain

$$L(W_K, W_Q, W_V, h) \tag{94}$$

$$= C + \mathbb{E}_{\widetilde{D}, x_{n+1}}(\hat{w}_{\widetilde{D},\Sigma}^{\top} u_{n+1} - h^{\top} W_V G_{\widetilde{D}} W_K^{\top} W_{Q_{:,1:d}} \Sigma^{1/2} u_{n+1})^2 \tag{95}$$

$$= C + \mathbb{E}_{\widetilde{D}, x_{n+1}}(\hat{w}_{\widetilde{D},\Sigma}^{\top} u_{n+1} - (h')^{\top} W_V' H^{-1} H G_{\widetilde{D}}' H \tag{96}$$

$$H^{-1} (W_K')^{\top} W_{Q_{:,1:d}}' \Sigma^{-1/2} \Sigma^{1/2} u_{n+1})^2 \tag{97}$$

$$= C + \mathbb{E}_{\widetilde{D}, x_{n+1}}(\hat{w}_{\widetilde{D},\Sigma}^{\top} u_{n+1} - (h')^{\top} W_V' G_{\widetilde{D}}' (W_K')^{\top} W_{Q_{:,1:d}}' u_{n+1})^2 . \tag{98}$$

In other words, this is equal to the loss obtained by the transformer corresponding to $h', W_V', W_K', W_Q'$ when the data is from the distribution studied in Section 3. The above proof also shows that the transformer corresponding to $h, W_V, W_K, W_Q$ has the same output on $x_{n+1}$, as the transformer corresponding to $h', W_V', W_K', W_Q'$ does on $u_{n+1}$. By Theorem 1, the output of the latter transformer on $u_{n+1}$ is

$$(\eta U^{\top} \vec{y})^{\top} u_{n+1} = \eta \vec{y}^{\top} U u_{n+1} \tag{99}$$

$$= \eta \sum_{i=1}^{n} y_i u_i^{\top} u_{n+1} \tag{100}$$

$$= \eta \sum_{i=1}^{n} y_i (\Sigma^{-1/2} x_i)^{\top} \Sigma^{-1/2} x_{n+1} \tag{101}$$

$$= \eta \sum_{i=1}^{n} y_i x_i^{\top} \Sigma^{-1} x_{n+1} , \tag{102}$$

where the learning rate is

$$\eta = \frac{\mathbb{E}_{\widetilde{D}\sim\mathcal{T}}[\hat{w}_{\widetilde{D},\Sigma}^{\top}U^{\top}\vec{y}]}{\mathbb{E}_{\widetilde{D}\sim\mathcal{T}}[\vec{y}^{\top}UU^{\top}\vec{y}]} \tag{103}$$

$$= \frac{\mathbb{E}_{\widetilde{D}\sim\mathcal{T}}[\vec{y}^{\top}X(X^{\top}X+\sigma^2\Sigma)^{-1}\Sigma^{1/2}U^{\top}\vec{y}]}{\mathbb{E}_{\widetilde{D}\sim\mathcal{T}}[\vec{y}^{\top}UU^{\top}\vec{y}]} \qquad \text{(By definition of } \hat{w}_{\widetilde{D},\Sigma})$$

$$= \frac{\mathbb{E}_{\widetilde{D}\sim\mathcal{T}}[\vec{y}^{\top}X(X^{\top}X+\sigma^2\Sigma)^{-1}\Sigma^{1/2}\Sigma^{-1/2}X^{\top}\vec{y}]}{\mathbb{E}_{\widetilde{D}\sim\mathcal{T}}[\vec{y}^{\top}X\Sigma^{-1/2}\Sigma^{-1/2}X^{\top}\vec{y}]} \qquad \text{(B.c. } U = X\Sigma^{-1/2})$$

$$= \frac{\mathbb{E}_{\widetilde{D}\sim\mathcal{T}}[\vec{y}^{\top}X(X^{\top}X+\sigma^2\Sigma)^{-1}X^{\top}\vec{y}]}{\mathbb{E}_{\widetilde{D}\sim\mathcal{T}}[\vec{y}^{\top}X\Sigma^{-1}X^{\top}\vec{y}]}\,. \tag{104}$$

This completes the proof. $\qquad\square$

## E    MISSING PROOFS FROM SECTION 5

*Proof of Lemma 5.* We proceed similarly to the proof of Lemma 1. For convenience, fix $\widetilde{D}$ and consider the function

$$g(u) = \mathbb{E}_{x_{n+1},y_{n+1}}[(u \cdot x_{n+1} - y_{n+1})^2 \mid \widetilde{D}]\,. \tag{105}$$

Then, we have

$$\nabla_u g(u) = \mathbb{E}_{x_{n+1},y_{n+1}}[2(u \cdot x_{n+1} - y_{n+1})x_{n+1} \mid \widetilde{D}]\,. \tag{106}$$

Therefore, if $\overline{u}_{\widetilde{D}}$ is the minimizer of $g(u)$ (note that $\overline{u}_{\widetilde{D}}$ depends on $\widetilde{D}$ but not on $x_{n+1}$ and $y_{n+1}$), then for any $u \in \mathbb{R}^d$, we have

$$\mathbb{E}_{x_{n+1},y_{n+1}}[(\overline{u}_{\widetilde{D}} \cdot x_{n+1} - y_{n+1})(u \cdot x_{n+1} - \overline{u}_{\widetilde{D}} \cdot x_{n+1}) \mid \widetilde{D}] = (u - \overline{u}_{\widetilde{D}}) \cdot \nabla_u g(\overline{u}_{\widetilde{D}}) = 0\,, \tag{107}$$

meaning that for any $u \in \mathbb{R}^d$, and some $C_{\widetilde{D}}$ which only depends on $\widetilde{D}$ and is independent of $u$,

$$\mathbb{E}_{x_{n+1},y_{n+1}}[(u \cdot x_{n+1} - y_{n+1})^2 \mid \widetilde{D}] \tag{108}$$

$$= \mathbb{E}_{x_{n+1},y_{n+1}}[(u \cdot x_{n+1} - \overline{u}_{\widetilde{D}} \cdot x_{n+1} + \overline{u}_{\widetilde{D}} \cdot x_{n+1} - y_{n+1})^2 \mid \widetilde{D}] \tag{109}$$

$$= \mathbb{E}_{x_{n+1},y_{n+1}}[(u \cdot x_{n+1} - \overline{u}_{\widetilde{D}} \cdot x_{n+1})^2 \mid \widetilde{D}] \tag{110}$$

$$+ \mathbb{E}_{x_{n+1},y_{n+1}}[(\overline{u}_{\widetilde{D}} \cdot x_{n+1} - y_{n+1})^2 \mid \widetilde{D}] \tag{111}$$

$$+ 2\mathbb{E}_{x_{n+1},y_{n+1}}[(u \cdot x_{n+1} - \overline{u}_{\widetilde{D}} \cdot x_{n+1})(\overline{u}_{\widetilde{D}} \cdot x_{n+1} - y_{n+1}) \mid \widetilde{D}] \tag{112}$$

$$= \mathbb{E}_{x_{n+1},y_{n+1}}[(u \cdot x_{n+1} - \overline{u}_{\widetilde{D}} \cdot x_{n+1})^2 \mid \widetilde{D}] \tag{113}$$

$$+ \mathbb{E}_{x_{n+1},y_{n+1}}[(\overline{u}_{\widetilde{D}} \cdot x_{n+1} - y_{n+1})^2 \mid \widetilde{D}] \qquad \text{(By Equation (107))}$$

$$= \|u - \overline{u}_{\widetilde{D}}\|_2^2 + C_{\widetilde{D}}\,. \tag{114}$$

Here the last equality is because $x_{n+1} \sim \mathcal{N}(0, I_{d\times d})$, and because $\mathbb{E}_{x_{n+1},y_{n+1}}[(\overline{u}_{\widetilde{D}} \cdot x_{n+1} - y_{n+1})^2 \mid \widetilde{D}]$ is a constant that depends on $\widetilde{D}$ but not on $u$. We can apply this manipulation to the loss function:

$$L(w, M) = \mathbb{E}_{\widetilde{D}\sim\mathcal{T}}\left[\mathbb{E}_{x_{n+1},y_{n+1}}[(y_{n+1} - \hat{y}_{n+1})^2 \mid \widetilde{D}]\right] \tag{115}$$

$$= \mathbb{E}_{\widetilde{D}\sim\mathcal{T}}\left[\mathbb{E}_{x_{n+1},y_{n+1}}[(y_{n+1} - w^{\top}G_{\widetilde{D}}Mv_{n+1})^2 \mid \widetilde{D}]\right] \tag{116}$$

$$= \mathbb{E}_{\widetilde{D}\sim\mathcal{T}}\left[\mathbb{E}_{x_{n+1},y_{n+1}}[(y_{n+1} - w^{\top}G_{\widetilde{D}}M_{:,1:d}x_{n+1})^2 \mid \widetilde{D}]\right] \tag{117}$$

$$= \mathbb{E}_{\widetilde{D}\sim\mathcal{T}}\left[\|\overline{u}_{\widetilde{D}} - M_{:,1:d}^{\top}G_{\widetilde{D}}w\|_2^2 + C_{\widetilde{D}}\right] \qquad \text{(By Equation (114))}$$

$$= \mathbb{E}_{\widetilde{D}\sim\mathcal{T}}[\|\overline{u}_{\widetilde{D}} - M_{:,1:d}^{\top}G_{\widetilde{D}}w\|_2^2] + C\,. \tag{118}$$

Here, $C$ is a constant independent of $w$ and $M$. This completes the proof of the lemma. $\qquad\square$

*Proof of Lemma 7.* We prove this using Assumption 1, imitating the proof of Lemma 3. For convenience, let $M(X) = \mathbb{E}[\vec{y}\vec{y}^\top \mid X]$. Then, the $(i,j)$-th entry of $M(X)$ is $\mathbb{E}_f[f(x_i)f(x_j)] + \sigma^2$ if $i = j$ and $\mathbb{E}_f[f(x_i)f(x_j)]$ if $i \neq j$, since $\mathbb{E}[\epsilon_i] = 0$ and the $\epsilon_i$ are i.i.d. and independent of $X$. If we perform the change of variables $x_i \to Rx_i$ for a fixed rotation matrix $R$ and all $i \in [n]$, then by Assumption 1, $\mathbb{E}_f[f(Rx_i)f(Rx_j)] = \mathbb{E}_f[f(x_i)f(x_j)]$, meaning that for any rotation matrix $R$,

$$\mathbb{E}_X[X^\top M(X)X] = \mathbb{E}_X[(XR^\top)^\top M(XR^\top)(XR^\top)] \tag{119}$$

$$= R\mathbb{E}_X[X^\top M(XR^\top)X]R^\top \tag{120}$$

$$= R\mathbb{E}_X[X^\top M(X)X]R^\top \tag{121}$$

by the rotational invariance of the $x_i$. This implies that $\mathbb{E}_X[X^\top M(X)X] = \mathbb{E}_{\widetilde{D}}[X^\top \vec{y}\vec{y}^\top X]$ is a scalar multiple of the identity matrix.

Next, we consider $\mathbb{E}_{\widetilde{D}\sim\mathcal{T}}[X^\top \vec{y}\vec{u}_{\widetilde{D}}^\top]$. For convenience, let $J(X) = \mathbb{E}_{\widetilde{D}\sim\mathcal{T}}[\vec{y}\vec{u}_{\widetilde{D}}^\top \mid X]$. If we make the change of variables $x_i \to Rx_i$, then the joint distribution of $y_1, \ldots, y_n, y_{n+1}$ does not change by Assumption 1, meaning that $\overline{u}_{\widetilde{D}}$ is replaced by $R\overline{u}_{\widetilde{D}}$. Thus, we can conclude that $J(X)$ is equivariant to rotations of all the $x_i$ by $R$:

$$J(XR^\top) = \mathbb{E}[\vec{y}\vec{u}_{\widetilde{D}}^\top \mid XR^\top] = \mathbb{E}[\vec{y}\vec{u}_{\widetilde{D}}^\top R^\top \mid X] = J(X)R^\top. \tag{122}$$

Thus,

$$\mathbb{E}_{\widetilde{D}\sim\mathcal{T}}[X^\top \vec{y}\vec{u}_{\widetilde{D}}] = \mathbb{E}_X[X^\top J(X)] \tag{123}$$

$$= \mathbb{E}_X[(XR^\top)^\top J(XR^\top)] \qquad \text{(By rotational invariance of } \mathcal{N}(0, I_{d\times d}))$$

$$= R\mathbb{E}_X[X^\top J(X)]R^\top \qquad \text{(By Equation (122))}$$

$$= R\mathbb{E}_{\widetilde{D}\sim\mathcal{T}}[X^\top \vec{y}\vec{u}_{\widetilde{D}}^\top]R^\top. \tag{124}$$

Thus, $\mathbb{E}_{\widetilde{D}\sim\mathcal{T}}[X^\top \vec{y}\vec{u}_{\widetilde{D}}^\top]$ is a scalar multiple of the identity matrix. The final statement of the lemma follows by taking the trace of the left and right hand sides. $\square$

*Proof of Lemma 8.* This follows by the same argument as Lemma 2 — here we use Lemma 6 in order to show that we only need to consider the lower left and upper right blocks of $G_{\widetilde{D}}$, and the rest of the proof follows from linear algebraic manipulations and applying Lemma 7 (in place of Lemma 4 which was used in the proof of Lemma 2). $\square$

*Proof of Theorem 3.* This follows directly from Lemma 8, since the effective linear predictor being $\eta X^\top \vec{y}$ is a necessary and sufficient condition for minimizing the pre-training loss. $\square$

# F    LEARNING RATE SIMULATIONS

We follow a setup similar to von Oswald et al. (2022). Our goal is to verify that a trained linear self-attention layer will implement a single step of gradient descent (GD), with a learning rate which matches our theoretical prediction in Theorem 1. As in von Oswald et al. (2022), we compare the predictions of our trained linear self-attention layer with 1 step of GD with learning rate $\eta$. One important difference between our experiment and that of von Oswald, et al. is that we set $\eta$ to be our theoretical prediction from Theorem 1, rather than selecting $\eta$ using line search. As in von Oswald et al. (2022), we evaluate the trained linear self-attention layer and the GD baseline on inputs which are scaled by a large amount compared to during training.

In our setting, the $x_i$'s are 10 dimensional vectors chosen so that each entry is uniformly random in $[-1, 1]$ (we note that this is similar to the setting of von Oswald et al. (2022) but different from our theoretical setting). The weight vectors $w$ for each sequence are random Gaussian vectors, chosen independently of the $x_i$'s. (Thus, if we scale the $x_i$'s by a certain $\alpha$ at evaluation time, then this leaves the distribution of $w$ unchanged, but the $y_i$'s are also scaled by a factor of $\alpha$ at test time compared to during training.) In both the training and the evaluation data, $\sigma^2 = 0.5$.

We estimate the learning rate for the GD baseline as follows. We predict $\eta = \frac{\mathbb{E}_{\widetilde{D}\sim\mathcal{T}}[\hat{w}_{\widetilde{D}}^\top X^\top \vec{y}]}{\mathbb{E}_{\widetilde{D}\sim\mathcal{T}}[\vec{y}^\top XX^\top \vec{y}]}$ as in Theorem 1. We then sample 100,000 sequences in order to estimate the expectations in each of the numerator and the denominator.

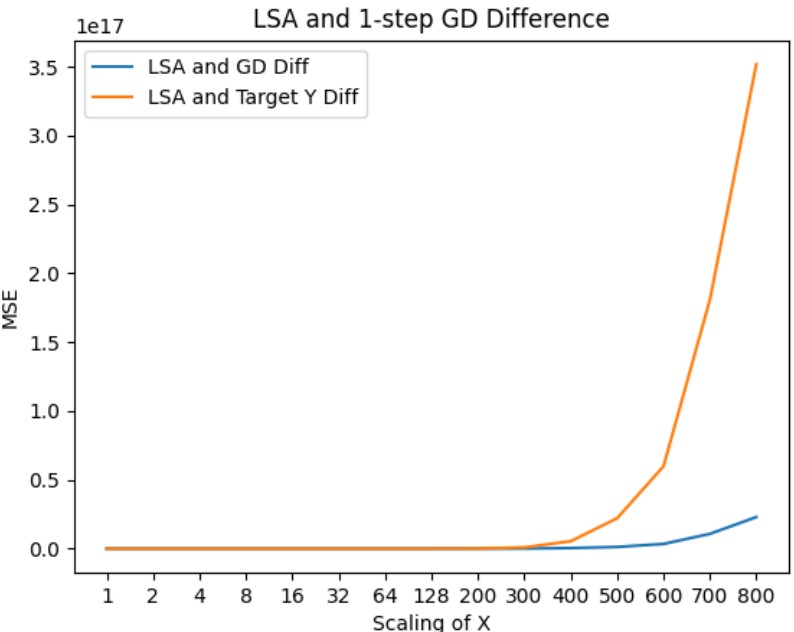

Figure 1: As the $x_i$ are scaled to be larger, the difference between the linear self-attention layer and the ground truth increases much more than the difference between the linear self-attention layer and the GD baseline.

Since the $x_i$'s are 10-dimensional vectors, $\begin{bmatrix} x_i \\ y_i \end{bmatrix}$ will be an 11-dimensional vector. Thus, to match our theoretical setting, the key and query dimensions of our linear self-attention are 11 as well. We train the linear self-attention layer using Adam, with a learning rate of 1e-4. We use a batch size of 2048 and train for 12000 steps. We also perform gradient clipping with a maximum norm of 10. In all sequences (at both train and test time), there are 10 context vectors $\begin{bmatrix} x_i \\ y_i \end{bmatrix}$, followed by one query vector $\begin{bmatrix} x \\ 0 \end{bmatrix}$.

See Figure 1 for the results. Our experiment shows that, as the inputs are scaled to be larger, the trained linear self-attention layer becomes farther from the ground truth, while the difference between the linear self-attention layer and the GD baseline does not grow as quickly, which suggests that the GD baseline with our predicted $\eta$ matches the algorithm that the trained linear self-attention layer learns. For instance, in the case where the scaling of the $x_i$ is 1, i.e. we do not rescale the $x_i$, the mean-squared error between the linear self-attention layer and the ground truth is about 2.6, while the difference between the linear self-attention layer and the GD baseline is about 0.14. Additionally, Figure 2 shows that as $\sigma^2$ increases (shown in the x-axis) the theoretically predicted learning rate (shown in the y-axis) decreases.

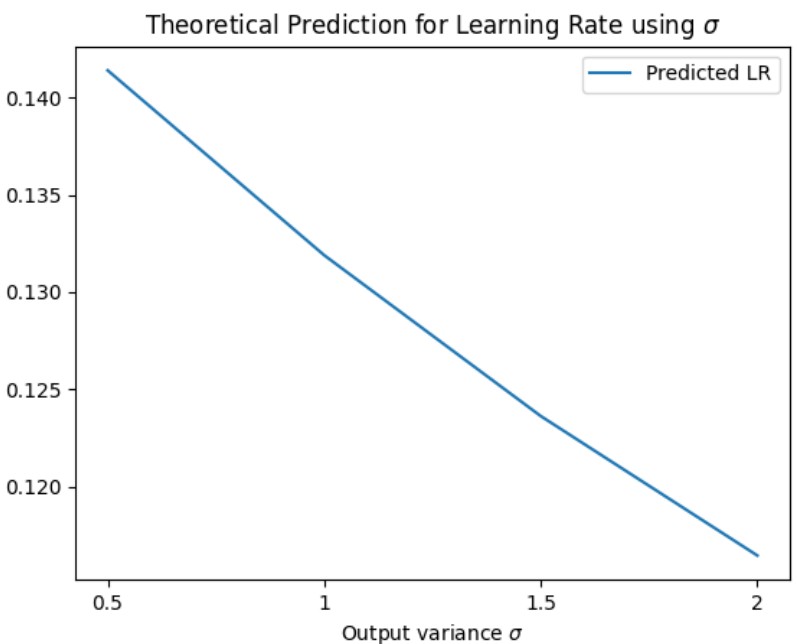

Figure 2: Change in learning rate as we vary $\sigma^2$.

