# OpenReview forum: "One Step of Gradient Descent is Provably the Optimal In-Context Learner with One Layer of Linear Self-Attention"
_ICLR.cc/2024/Conference — ICLR 2024 poster_

### Official Review · Reviewer_nLRu · 2023-10-31

**Soundness:** 2 fair
**Presentation:** 3 good
**Contribution:** 2 fair
**Rating:** 5
**Confidence:** 4

**Summary:**

This paper theoretically studies how one-layer transformers with linear attention implement one step of gradient descent on least-squares linear regression. The results include the cases when the covariates are from a standard Gaussian, a  non-standard Gaussian, and when the target function is nonlinear. The conclusion covers the global convergence of the network.


---------------------------------------------------------------------

After rebuttal, I tend to maintain the score of 5. The main concern is the significance to the community.

**Practical Insight**: It is still a major concern. I am not clear on what I can learn from this paper. It makes the result less interesting and significant to me. For example, can this paper provide an explanation for any phenomenon in in-context learning in practice? How can this paper guide the training in practice?

**Experiments**: Generally, I am satisfied with the efforts of the authors. Since I only specified one experiment, I will not treat the experiment part as a big weakness, although I expect to see more experiments.

**Fully connected neural networks and Assumption 1**: Good, I am satisfied with this result.

**Contribution compared to [Zhang, et al. 2023]**: OK. Although [Zhang et al., 2023] was posted online 3.5 months before ICLR submission deadline, I agree it can be treated as a concurrent work.

**Strengths:**

The significance is good since the studied problem is essential and interesting to the community. The paper is overall well-written with good clarity. This paper provides a comparison with existing works and concurrent works. The contributions include that it provides a global optimal analysis when constructing a linear-attention Transformer to implement gradient descent. Meanwhile, it shows analyses on non-standard Gaussian inputs and non-linear target functions.

**Weaknesses:**

1. This paper lacks empirical justification.
2. I am not sure about the practical insight from the theoretical analysis of this work.

**Questions:**

1. Can you verify that the $\eta$ in Theorem 1,2 are as predicted by experiments? Specifically, can you show how $\sigma^2$ in Theorem 2 affects $\eta$ by experiments?

2. I don't know why fully-connected neural networks satisfy Assumption 1 (1). Can you provide a proof for this claim?

3. Without section 5, the contribution compared with Zhang el at., 2023 will only be incremental. Why do you assign too much content to Section 3? I think it is better to enlarge the content of Section 5.

[Zhang et al., 2023] " Trained transformers learn linear models incontext."

---

> ### Author Response · Authors · 2023-11-23
>
> We thank the reviewer for their feedback, and respond to the comments below:
>
> **Practical Insight:** Our work makes progress towards understanding the mechanism by which transformers perform in-context learning. Theoretical analyses usually need to start with simple cases to build intuition. Our setting with i.i.d. examples $(x_i, y_i)$ is also motivated by the few-shot in-context learning setting seen in practice.
>
> **Experiments:** Please see Appendix D (newly added during discussion period), where we give experimental results with the goal of showing that the linear self-attention layer implements 1 step of GD, with learning rate matching our theoretical prediction. We observe that the learning rate decreases as sigma increases.
>
> **Fully-connected neural networks (NNs) and Assumption 1:** We can show that fully-connected NNs satisfy this assumption as follows. For simplicity, consider a two-layer neural network $f(x) = W_2 \sigma(W_1 x)$ where $\sigma$ is a nonlinearity, and $W_1$, $W_2$ have i.i.d. Gaussian entries. Let $R$ be a fixed rotation matrix. Then, $(f \circ R)(x) = W_2 \sigma (W_1 R x)$. Since $R$ is a fixed rotation matrix and $W_1$ is a Gaussian matrix, $W_1R$ has the same distribution as $W_1$. Thus, the random function $f \circ R$ has the same distribution as $f$.
>
> **Contribution relative to Zhang, et al. 2023:** We note that Zhang, et al. 2023 is a concurrent/independent work, which was posted on arxiv around the same time as ours.

---

### Official Review · Reviewer_1cqQ · 2023-11-01

**Soundness:** 4 excellent
**Presentation:** 4 excellent
**Contribution:** 3 good
**Rating:** 8
**Confidence:** 4

**Summary:**

In this paper, the authors provide a theoretical analysis of transformers equipped with a single layer of linear self-attention, trained on synthetic noisy linear regression data. The primary focus of this paper lies in exploring in-context learning pretraining scenarios, where the training data consists of pairs (x_i, y_i) with associated ground truth, and the evaluation is based on the Mean Squared Error (MSE) metric for the test point (x', y').

The key findings presented in this paper can be summarized as follows: (1) Under the assumption of linear noisy ground truth, when x_i samples are drawn from an isotropic Gaussian distribution, the one-layer transformer model that minimizes the pretraining loss effectively corresponds to a single step of Gradient Descent (GD) applied to the least-squares linear regression problem. (2) When x_i samples are drawn from a non-isotropic Gaussian distribution, the optimization process becomes a preconditioned GD. The authors shed light on this aspect, showcasing the connection between the nature of the input distribution and the optimization approach. Furthermore, The paper goes beyond linear cases, demonstrating that the findings can be extended to non-linear scenarios under specific symmetric conditions.

In conclusion, I strongly recommend accepting this paper for the following reasons: (1) The paper demonstrates exceptional organization, making it highly accessible and comprehensible for the readers. (2) The topic addressed in this paper holds paramount significance within the Language Model (LLM) domain, contributing to our understanding of key theoretical aspects. (3) The paper introduces some innovative results, particularly in the sections related to preconditioning and non-linear extensions. These novel findings are likely to ignite further research and inspire intriguing follow-up studies.

**Strengths:**

Overall, this paper has the potential to inspire and stimulate further research in this area.
1. The organization of the paper is well-structured, making it accessible to a broad readership.
2. The paper addresses a crucial topic in the realm of Language Model (LLM) research, shedding theoretical insights on transformers under in-context learning scenarios.
3. The results presented in the paper are noteworthy, particularly the connections made in Theorem 1, including the proof of global minimization and its equivalence to a single step of gradient descent. The exploration of non-isotropic Gaussian distributions leading to preconditioned GD is an interesting and novel aspect. Additionally, the extension to non-linear cases adds depth to the research.
4. The paper is well-written and effectively communicates its findings and insights.

**Weaknesses:**

While the paper is commendable, there are a couple of minor questions and potential areas for further investigation:

1. The usage of the statement "(Wk, Wq, Wv, h) is a global minimizer" in Theorem 1 raises questions about the specifics of this minimization process. Further clarification or details regarding this construction might be beneficial for readers.

2. The reviewer suggests that, in in-context learning regimes, the downstream phase is crucial. Encouraging future research that delves into this aspect could be valuable for a more comprehensive understanding of the subject.

**Questions:**

See weakness.

---

> ### Author Response · Authors · 2023-11-23
>
> We thank the reviewer for their positive review, and for mentioning that the results are noteworthy.
>
> **Clarification on global minimizers:** We note that all global minimizers must implement the same linear predictor, by Lemma 2. Thus, while (Wk, Wq, Wv, h) is not unique, the function which the resulting transformer implements is unique. We will include this clarification in the final version.

---

### Official Review · Reviewer_VQYN · 2023-11-05

**Soundness:** 2 fair
**Presentation:** 2 fair
**Contribution:** 2 fair
**Rating:** 5
**Confidence:** 3

**Summary:**

This paper considers a one layer self-attention model with linear attention and shows that one step of gradient descent is the optimal in-context learner in this case. Specifically, they consider a synthetic noisy linear regression task and show that when the covariates are drawn from a standard Gaussian, the model implements one step of GD, which is also the global minimizer of the pretraining loss. If the distribution of the covariates if changed to a non-isotropic Gaussian, it now implements pre-conditioned GD. On the other hand, when using a nonlinear model to generate the data, it still implements a single step of GD.

**Strengths:**

This paper takes a step to improve the theoretical understanding of in-context learning in transformers, which is an important topic.

**Weaknesses:**

While this is an important topic, the paper does not seem to make a significant contribution. The main drawback is that it considers a one layer attention model, which has been studied extensively for the developing theoretical understanding of in-context learning.

In the first case, the only contribution seems to be that using an appropriate step size allows the resulting solution to be a global minimizer of the pretraining loss. This does not seem to add to the understanding of transformers, as it was already shown in [1] that transformers implement one step of GD. Similarly, the result in the third case is also not very informative. Given that this is a one layer model, it is not surprising that it implements one step of GD, even when the target function is nonlinear.

[1] von Oswald et al. 'Transformers learn in-context by gradient descent', ICML 2023.

**Questions:**

Please see the weaknesses section. My main concern is that this paper does not offer new insights regarding in-context learning in transformers (that has not been discussed in one of the prior works), and also does not use any new proof techniques. It would be interesting to analyze multi-head attention or multilayer transformers, as the authors discuss in the conclusion.

---

> ### Author Response · Authors · 2023-11-23
>
> We thank the reviewer for their comments. We emphasize that all the previous works on this topic, including [1], are empirical. von Oswald et al. empirically show that a 1-layer transformer with linear self-attention trained with GD will implement 1 step of gradient descent, while our contribution is to give a theoretical proof.
>
> We note that in the case of a nonlinear target function, while it is clear that the 1-layer transformer will always implement a linear model, it is not necessarily clear that it should implement 1 step of gradient descent.

---

### Official Review · Reviewer_ksN8 · 2023-11-07

**Soundness:** 3 good
**Presentation:** 3 good
**Contribution:** 3 good
**Rating:** 6
**Confidence:** 4

**Summary:**

This paper theoretically analyzed the one-layer linear self-attention layer on the linear regression teacher model. The authors proved that after pertaining to this one-layer transformer under square loss, the minimizer we got is equivalent to a single-step gradient descent (GD) on the least-squares linear regression problem. This paper also considered covariate shifts for the data distribution, which correspond to preconditioned GD. Finally, the authors claimed for rotational invariant nonlinear teacher models, the global minimizer of the transformer is still equivalent to one step GD on least-squares linear regression.

**Strengths:**

The paper is written very clearly in a way that highlights the main analysis techniques in the main body. It also provides enough summaries for some concurrent works and literature reviews. The main message of this paper is clear. The authors theoretically analyze the in-context learning capability for the self-attention layer in the linear regime and make the connection with the Bayes-optimal predictor of the linear regression model. This result provides a number of natural directions for future theoretical study in transformers.

**Weaknesses:**

1. The dataset assumption is simple and the authors only considered i.i.d. data sequence with linear teacher model. This setting helps the analysis but may not be able to fully capture the properties of the self-attention layer. Besides, the proofs rely on the rotational invariance of Gaussian distribution. It would be interesting to generalize the results in non-Gaussian datasets or consider more dependent structures in the data sequence, like the Bigram language model in [1].

2. This paper focuses on the global minimizer of the population square loss of the self-attention layer which simplifies the analysis. It would be natural to consider the minimizer of the empirical loss during the pre-training process and how the minimizer of the GD or stochastic GD with finite step sizes generalizes in the test point.

3. Further experiments and simulations should be presented for completeness. For instance, the training dynamic of nonlinear/multi-layer transformers with nonlinear target functions that are defined in Section 5. This will help us know the limitations of the current theory and potential interesting directions for future analysis.

**Questions:**

1. You may need to briefly explain the parameter $\eta$ in Eq. (1).

2. In the second paragraph on page 4, $v_n=\\begin{bmatrix}x_i
\\\\ 0 \\end{bmatrix}$ should be $v_n=\\begin{bmatrix}x_{n+1}
\\\\ 0 \\end{bmatrix}$.

3. In Eq. (10), the number of training parameters is $d^2+d$ and we consider population squared loss for training. Does that mean this model is under-parameterized and has a unique global minimizer? And when could the constants in Lemma 1 and Eq. (13) in Lemma 2 be zero? More specifically, is the minimizer constructed in Theorem 1 unique and when will it attain zero training loss?

4. How large the learning rate $\eta$ is? Following the remark after Theorem 1, we know the global minimizer is equivalent to a step gradient descent on the empirical loss of the least squares problem with zero initialization and learning rate $\eta$ defined in Theorem 1. How large $\eta$ is, compared with the largest eigenvalue of the Hessian matrix $H$ of this least squares problem? Is it just close to or larger than the maximal learning rate $2/\lambda_{\max}(H)$?

5. In Section 4, when we consider data covariance in $\Sigma$, why do we renormalize back by $w\sim\mathcal{N}(0,\Sigma^{-1})$? Can we consider $w$ has another different population covariance like [2]?

6. In Theorem 3, when defining $\eta$, what is $\mathcal{D}$? No definition of this distribution.

7. In the proof of Lemma 1, after Eq. (29), why is the minimizer of $g(u)$ given by $\hat{w}_{\tilde{D}}$? Here, for $g(u)$, you only have one data point. Can you explain more?

8. It may be worthy to mention or compare with some of the references among [1] and [3-7].


=================================================================================================

[1] Bietti, et al. "Birth of a Transformer: A Memory Viewpoint."

[2] Wu and Xu. "On the Optimal Weighted $\ell_2 $ Regularization in Overparameterized Linear Regression."

[3] Takakura and Suzuki. "Approximation and Estimation Ability of Transformers for Sequence-to-Sequence Functions with Infinite Dimensional Input."

[4] Tarzanagh, et al. "Margin Maximization in Attention Mechanism."

[5] Tarzanagh, et al. "Transformers as support vector machines."

[6] Bai, et al. "Transformers as Statisticians: Provable In-Context Learning with In-Context Algorithm Selection."

[7] Guo, et al. "How Do Transformers Learn In-Context Beyond Simple Functions? A Case Study on Learning with Representations."

---

> ### Author Response · Authors · 2023-11-23
>
> We thank the reviewer for their detailed review. Below we respond to each of the comments.
>
> **The dataset assumption is simple:** Theoretical analyses usually need to start with simple cases. Compared to the bigram language model setting of [1], our setting with i.i.d. examples $(x_i, y_i)$ is closer to the few-shot in-context learning setting seen in practice. Also, many previous theoretical works analyze deep linear neural networks to obtain intuition, such as Gunasekar, et al. (2018).
>
> **Consider the minimizer of the empirical loss, how the minimizer of GD generalizes:** We believe that our analysis of the global minimizer of the population loss is already an interesting finding. We also note that understanding the population loss is, in general, a big step towards understanding the empirical loss, as standard concentration bounds can be applied to bound the difference between the population and empirical losses, given enough samples. There are also several existing works which study the implicit regularization of gradient descent on linear or non-linear neural networks, such Gunasekar, et al. (2018) or Damian, et al. (2021). We consider the minimizer of the empirical loss or the generalization ability of GD to be orthogonal questions for our work.
>
> Damian, et al. (2021) Label Noise SGD Provably Prefers Flat Global Minimizers
> Gunasekar, et al. (2018) Implicit Bias of Gradient Descent on Linear Convolutional Networks
>
> **Experiments:** Please see appendix D (newly added during discussion phase), where we include some preliminary experiments with the goal of confirming that a trained linear self-attention layer learns to implement 1 step of GD with our theoretically predicted learning rate.
>
> **Is the global minimizer unique?:** The global minimizer is essentially unique (up to rescaling of $w, M$, etc.) since as we show in Lemma 2, any global minimizer of the population loss must implement the linear predictor whose weight vector is $\eta X^\top y$. However, this minimizer will not attain zero population loss, due to the output noise. Furthermore, the linear predictor which minimizes the Bayes risk is given by ridge regression, but finding the optimal weight vector for ridge regression requires performing matrix inversion, which cannot be represented by a single linear self-attention layer.
>
> **Learning rate v.s. Hessian singular value:** The learning rate $\eta$ is roughly the same size as $2/\lambda$ - we will include a proof in the final version. We can simplify the numerator and denominator of $\eta$ using the equality $E[yy^T | X] = XX^T + \sigma^2 I$, as well as the observation from this link https://stats.stackexchange.com/questions/589669/gaussian-fourth-moment-formulas, to find that the learning rate is $\approx 1/(N + d)$, assuming that the output noise variance $\sigma^2$ is very small. Meanwhile, the Hessian of the least-squares problem is $X^T X$, which has a maximum singular value which is $O((\sqrt{d} + \sqrt{N})^2) = O(d + N)$.
>
> **Different Population Covariance:** We believe our analysis in Section 4 can extend to the case where the weight vector w has the identity covariance.
>
> **Minimizer of $g(u)$:** Here we are using the fact mentioned in e.g. Section 4.3 of Akyurek, et al. (2022) that the ridge regression weight vector is the linear predictor which minimizes the Bayes risk in the noisy linear regression setting.
>
> Akyurek, et al. (2022). What learning algorithm is in-context learning? Investigations with linear models
>
> **Comparison with related works:** The work of Takakura and Suzuki theoretically shows that transformers can achieve low approximation error when the target function is shift-equivariant on sequences of infinite length, subject to certain regularity conditions. Bai, et al. propose various constructions through which transformers can solve problems such as Bayesian linear regression and generalized linear models, achieving low test error - they also show that transformers can represent a form of model selection. Guo, et al. similarly provide constructions of transformers which can represent certain algorithms. Compared to these works which give approximation-theoretic constructions, we show that one step of gradient descent is the best predictor that can be implemented by 1 layer of linear self-attention. Interestingly, it follows from our Lemma 2 that this is the unique linear predictor that can be implemented by 1 layer of linear self-attention which minimizes the population loss.
>
> Tarzanagh, et al. [4] analyze the global minima and training dynamics of transformer layers with general data. They show that the global minimum corresponds with a type of max-margin solution, and give conditions under which the training dynamics converge to this global minimum. We note however that this work was released on arXiv at around the same time as ours, and was published in NeurIPS only shortly before the ICLR submission deadline.

---

### Meta-Review · Area_Chair_3QeU · 2023-12-13

**Metareview:**

This paper studies the global minimizers of in-context learning of linear regression in a one-layer transformer with linear attention. It shows that for isotropic Gaussian covariates, the optimal solution is one step of gradient descent. For non-isotropic Gaussian, this becomes one step of preconditioned gradient descent. It is further shown that even for nonlinear responses, the global optimum is still one-step gradient descent on a linear regression objective.

Understanding in-context learning is a major open question in language models, and a recent line of work started to look into this question in simple synthetic problems like linear regression. This paper makes a solid contribution in this direction by studying what are the global optima of the pretraining loss. The main concerns raised by the reviewers are that the paper focuses on a simple and idealistic setting and that the paper does not have a practical implication.

**Justification For Why Not Higher Score:**

While the paper makes solid theoretical contributions, the impact on practitioners might not be significant.

**Justification For Why Not Lower Score:**

Solid results worth sharing with the community.

---

### Decision · Program_Chairs · 2024-01-16

Accept (poster)